# Glycan cross-feeding supports mutualism between *Fusobacterium* and the vaginal microbiota

**Kavita Agarwal**[1,2¤], **Lloyd S. Robinson**[1,2], **Somya Aggarwal**[1,2], **Lynne R. Foster**[1,2], **Ariel Hernandez-Leyva**[2,3], **Hueylie Lin**[1,2], **Brett A. Tortelli**[2,4], **Valerie P. O'Brien**[1,2], **Liza Miller**[1,2], **Andrew L. Kau**[2,3], **Hilary Reno**[5], **Nicole M. Gilbert**[2,6,7], **Warren G. Lewis**[1,2¤]*, **Amanda L. Lewis**[1,2,6¤]*

**1** Department of Molecular Microbiology, Washington University School of Medicine, St. Louis, Missouri, United States of America, **2** Center for Women's Infectious Disease Research, Washington University School of Medicine, St. Louis, Missouri, United States of America, **3** Division of Allergy and Immunology, Department of Medicine, Washington University School of Medicine, St. Louis, Missouri, United States of America, **4** Department of Genetics, Washington University School of Medicine, St. Louis, Missouri, United States of America, **5** Division of Infectious Disease, Department of Medicine, Washington University School of Medicine, St. Louis, Missouri, United States of America, **6** Department of Obstetrics and Gynecology, Washington University School of Medicine, St. Louis, Missouri, United States of America, **7** Center for Reproductive Health Sciences, Washington University School of Medicine, St. Louis, Missouri, United States of America

¤ Current address: Department of Obstetrics, Gynecology, and Reproductive Sciences, Glycobiology Research and Training Center, University of California San Diego, La Jolla, California, United States of America

* a1lewis@ucsd.edu (ALL); WarrenGLewisPhD@gmail.com (WGL)

**Data Availability Statement:** All relevant data are within the paper and its Supporting Information files.

## Abstract

Women with bacterial vaginosis (BV), an imbalance of the vaginal microbiome, are more likely to be colonized by potential pathogens such as *Fusobacterium nucleatum*, a bacterium linked with intrauterine infection and preterm birth. However, the conditions and mechanisms supporting pathogen colonization during vaginal dysbiosis remain obscure. We demonstrate that sialidase activity, a diagnostic feature of BV, promoted *F. nucleatum* foraging and growth on mammalian sialoglycans, a nutrient resource that was otherwise inaccessible because of the lack of endogenous *F. nucleatum* sialidase. In mice with sialidase-producing vaginal microbiotas, mutant *F. nucleatum* unable to consume sialic acids was impaired in vaginal colonization. These experiments in mice also led to the discovery that *F. nucleatum* may also "give back" to the community by reinforcing sialidase activity, a biochemical feature of human dysbiosis. Using human vaginal bacterial communities, we show that *F. nucleatum* supported robust outgrowth of *Gardnerella vaginalis*, a major sialidase producer and one of the most abundant organisms in BV. These results illustrate that mutually beneficial relationships between vaginal bacteria support pathogen colonization and may help maintain features of dysbiosis. These findings challenge the simplistic dogma that the mere absence of "healthy" lactobacilli is the sole mechanism that creates a permissive environment for pathogens during vaginal dysbiosis. Given the ubiquity of *F. nucleatum* in the human mouth, these studies also suggest a possible mechanism underlying links between vaginal dysbiosis and oral sex.

**Funding:** This work was funded by the National Institute of Allergy and Infectious Diseases (R01 AI114635 to ALL and WGL, R01 AI127554 to WGL, and K08 AI113184 to ALK), the National Institute of Diabetes and Digestive and Kidney Diseases (K01 DK110225 to NMG), the Eunice Kennedy Shriver National Institute of Child Health and Human Development (F30HD094435 to BAT), the Burroughs Wellcome Fund Preterm Birth Initiative (to ALL), a pilot grant from March of Dimes Prematurity Research Center at Washington University (to AL), and a Morse Postdoctoral Fellowship (to KA). The funders had no role in the study design; collection, analysis, and interpretation of data; writing of the paper; and/or decision to submit for publication.

**Competing interests:** The authors declare that they have no relevant conflicts of interest.

**Abbreviations:** ATCC, American Type Culture Collection; BV, bacterial vaginosis; *catP*, chloramphenicol acetyltransferase; DMB, 1,2-diamino-4,5-methylenedioxybenzene; dpi, days postinoculation; hpi, hours postinoculation; HPLC, high-performance liquid chromatography; IP, intraperitoneal; LB, lysogeny broth; LOD, limit of detection; MWCO, molecular weight cutoff; NanA, *N*-acetylneuraminate lyase; Neu5Ac, *N*-acetylneuraminic acid; NR, nonredundant; OD, optical density; PBS, phosphate-buffered saline; RT-qPCR, reverse transcription–quantitative PCR; *siaT*, predicted sialic acid transporter; Sm$^R$, streptomycin-resistant; *tuf*, Translation elongation factor Tu; UC, Urogenital Community; WT, wild type; 3SL, 3′-sialyllactose; 4MU, 4-methylumbelliferone.

## Introduction

Bacterial vaginosis (BV) is an imbalance or dysbiosis of the vaginal microbiome that affects approximately 29% of US women [1]. BV is characterized by low levels of "beneficial" lactobacilli and overgrowth of bacteria from diverse taxonomic groups [2–4]. Women with BV are at higher risks of various adverse health outcomes and infection-related complications and are more likely to be vaginally colonized by the potentially pathogenic bacterial taxa that cause these infections [5–12]. For example, BV-positive women and those with amniotic fluid infections and preterm labor are more likely to be vaginally colonized by *Fusobacterium nucleatum* [3,9,13]. Unfortunately, mechanisms of pathogen colonization within the BV niche are poorly understood. It is widely held that a healthy microbiome thwarts the entry of potentially pathogenic members into the community (referred to as "colonization resistance") and that the loss of this "colonization resistance" plays a vital role in the development of vaginal dysbiosis [14]. We suggest that bacterial communities of the dysbiotic vaginal niche also rely on mutually supporting metabolic interactions to stabilize communities, as described in other microbial ecosystems [15–19].

One biochemical characteristic of BV is the presence of sialidase activity in vaginal fluids [20–22]. Sialidase activity may contribute to complications of BV. For example, high sialidase activity in vaginal fluids is associated with increased risk of preterm birth [21,23]. Sialidases play diverse roles in bacterial–host interactions, coinfection, and dysbiosis in oral, gastrointestinal, and airway systems [17,18,24–30]. We hypothesize that bacterial sialidases also play important roles in pathogen colonization and dysbiosis in the vagina. Bacterial sialidases liberate mucosal carbohydrates called sialic acids from glycan chains of secreted mucus components and cell surface glycoproteins. In BV, vaginal mucus secretions have higher levels of free/liberated sialic acid [31–33]. *Gardnerella vaginalis*, an abundant species in BV, produces sialidases to release and forage sialic acids from glycan chains of sialoglycoproteins [31,34]. However, not all bacteria that catabolize sialic acids express sialidase [35,36]. Here, we examine the idea that in the sialidase-positive vaginal niche, the ability to utilize free sialic acid may provide fitness benefits to sialidase-negative potential pathogens, facilitating their growth and persistence.

*F. nucleatum* is a gram-negative spindle-shaped obligate anaerobe that is ubiquitous in the human mouth but also one of the most commonly isolated microorganisms from amniotic fluid of women in preterm labor [37–39]. Although the prevailing view is that *F. nucleatum* reaches the amniotic fluid through an oral–hematogenous route [40], vaginal colonization with *F. nucleatum* is also a risk factor for amniotic fluid infection and preterm birth [13]. *F. nucleatum* is not known to express sialidase of its own [41], but it commonly resides among sialidase-producing bacteria in the mouth, gut, and vagina [42–46]. Some *F. nucleatum* strains also encode putative sialic acid catabolic pathways, although functional characterization of the corresponding gene products is still limited [47–51]. We hypothesized that *F. nucleatum* may derive nutritional benefit from host sialoglycans when exogenous sialidases produced by other bacteria are present. Here, we use in vitro approaches and a mouse model to show that *F. nucleatum* cannot obtain sialic acid from intact glycoconjugates, in which sialic acid is bound to underlying glycans, but can nevertheless benefit from sialic acid catabolism when colonizing a sialidase-positive vaginal niche. Unexpectedly, our models also led to the discovery that *F. nucleatum* does not act in a simple one-way relationship with sialidase-producing bacteria, but rather engages in a mutually beneficial relationship. In fact, the data demonstrate that *F. nucleatum* exposure to vaginal communities may encourage features of dysbiosis (increased sialidase activity and *G. vaginalis* abundance) in susceptible vaginal communities. These results may help explain why women with BV are at increased risk of vaginal colonization by some

pathogens such as *F. nucleatum*. Additionally, our data suggest that mutual reinforcement between bacterial species, made possible in part through metabolite cross-feeding, promotes pathogen colonization and vaginal dysbiosis.

## Results

### Subsets of *F. nucleatum* consume free sialic acid

An organism's ability to colonize an already-occupied niche depends on its ability to access nutrients for growth. Some mucosal bacteria forage on host sialic acids by importing them from the extracellular environment through a sialic acid transporter and then converting them to *N*-acetyl-mannosamine and pyruvate using a sialate lyase (*N*-acetylneuraminate lyase, NanA) [52–54]. Recent studies suggest that *F. nucleatum* may encode these functionalities [48–50]. Here, we used an amino acid sequence of the well-characterized *Escherichia coli* NanA (MG1655, GenBank: AAC76257.1) to search for homologous proteins in *F. nucleatum*. A bioinformatic analysis of 28 *F. nucleatum* proteomes spanning several subspecies identified NanA homologs in most members of the subspecies *nucleatum* and *vincentii* and some strains of *animalis*, but not in the subspecies *polymorphum* (S1 Fig). Functional analyses of free sialic acid foraging among a collection of *F. nucleatum* strains were consistent with these genomic predictions. Members of the subspecies *nucleatum* and *vincentii* were competent foragers of free sialic acid, but the tested strains of *animalis* and *polymorphum* were not (Fig 1A). Analyses of other members of the order Fusobacteriales suggested that some species in addition to *F. nucleatum* (for example, *F. mortiferum*) may catabolize sialic acids, but sialic acid consumption is not ubiquitous.

To test whether the *nanA* identified from *F. nucleatum* strain American Type Culture Collection (ATCC) 23726 encodes a functional sialic acid lyase, the gene was cloned into an *E. coli* expression plasmid and transformed into a *ΔnanA* deletion mutant of *E. coli* [36] (S1 Table). On its own, *E. coli ΔnanA* (containing an empty vector) was unable to grow on sialic acid as a sole carbon source (Fig 1B). However, growth was restored by complementation with plasmids containing either *E. coli nanA* or *F. nucleatum nanA*. Sialate lyase activity was also assayed in clarified lysates by measuring the disappearance of *N*-acetylneuraminic acid (Neu5Ac) over time. The amount of remaining sialic acid was measured by fluorescent derivatization and high-performance liquid chromatography (HPLC) analysis by comparison to synthetic Neu5Ac standards processed in parallel. As expected, the *E. coli ΔnanA* strain containing empty vector did not have sialic acid lyase activity (Fig 1C). However, in reactions with lysates from the complemented strains, >80% of the sialic acid was depleted within 30 min (Fig 1C). These results demonstrate that *F. nucleatum* NanA (GenBank: EFG95907.1) functions as a sialate lyase.

In *F. nucleatum* (ATCC23726), the *nanA* gene is found downstream of a putative sialic acid transporter (*siaT*) in a gene cluster encoding other enzymes with predicted roles in carbohydrate metabolism (S2 Table). To further assess the importance of *F. nucleatum* sialic acid catabolism, a minimal suicide vector targeting the *siaT* gene was designed for insertional mutagenesis in a streptomycin-resistant (Sm^R) isolate of *F. nucleatum* ATCC23726 (Fig 2A). This plasmid was used to generate a strain in which *siaT* was disrupted (Ω*siaT*; Fig 2B). Hereafter, the *F. nucleatum* wild type (WT) and Ω*siaT* mutant are referred to as "WT" and "Ω*siaT*." Analysis of the transcripts by reverse transcription–quantitative PCR (RT-qPCR) throughout this gene cluster in the Ω*siaT* mutant showed that the transcription of genes upstream of *siaT* were unaffected by the insertion (Fig 2C). As expected, there was an approximately 5-fold reduction in transcript of 3 genes downstream of the mutation site that are involved in sialic acid catabolism. Consistent with these findings, Ω*siaT* was unable to forage on free sialic acid

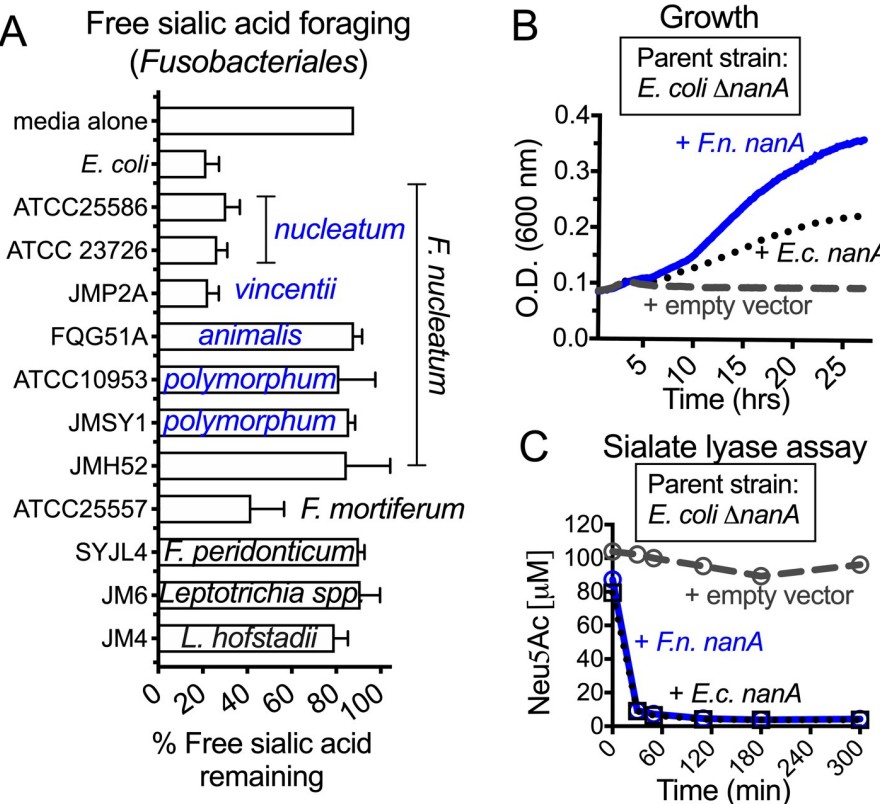

**Fig 1. Analysis of sialic acid utilization by Fusobacteria and functional evidence of *F. nucleatum*-encoded sialate lyase (*nanA*).** (A) Amount of sialic acid remaining in the growth medium 24–48 h postinoculation of different Fusobacteria (as indicated) or uninoculated media control. Subspecies of *F. nucleatum* are indicated in blue text. All Fusobacteria were grown in supplemented Columbia medium with free sialic acid added (Neu5Ac, 100 μM)—medium that was also used as uninoculated control. *E. coli* (MG1655), a known sialic acid consumer, cultured in LB medium with added free sialic acid (Neu5Ac, 100 μM) was included as a positive control. Error bars shown are standard deviation of the mean from 3 independent experiments. (B–C) *E. coli* MG1655 *ΔnanA* was complemented with empty vector, *E. coli nanA* or putative *nanA* from *F. nucleatum* ATCC23726 (GenBank accession EFG95907). Data shown are representative of 3 independent experiments. (B) Growth was assessed in minimal media with sialic acid by measuring absorbance at 600 nm. (C) Lysates of the *E. coli nanA* mutant with empty vector or the complemented strain were incubated with sialic acid (Neu5Ac), and its disappearance was monitored over time by fluorescent derivatization (with DMB) followed by HPLC. The underlying numerical data for this figure can be found in S1 Data. Strain names: FQG51A, JM6, JM4, JMH52, JMP2A, JMSY1, SYJL4. Description of bacterial strains can be found in S1 Table. ATCC, American Type Culture Collection; DMB, 1,2-diamino-4,5-methylenedioxybenzene; *E.c.*, *E. coli*; *F.n.*, *F. nucleatum*; HPLC, high-performance liquid chromatography; LB, lysogeny broth; *nanA*, *N*-acetylneuraminate lyase; Neu5Ac, *N*-acetylneuraminic acid; OD, optical density.

(Neu5Ac) (Fig 2D), even though both strains grew similarly in the nutrient-rich media used for this experiment (S2 Fig). *F. nucleatum* is a fastidious anaerobe, and single-carbon–source experiments are not feasible as an experimental strategy. Therefore, we developed a nutrient-limited media containing some of the base components of Columbia media, as described in the Methods. Reducing the nutrient content of the growth media made it possible to investigate the impact of sialic acid foraging on the growth of *F. nucleatum* when resources are limited. Experiments were performed in nutrient-limited media with or without added carbohydrate sources. Briefly, WT and Ω*siaT* were inoculated into a nutrient-limited medium, either with no added carbohydrate or with added monosaccharides—glucose or free sialic acid (Neu5Ac). Both WT and Ω*siaT* grew poorly in low-carbohydrate medium, reaching a plateau at an optical density (OD)$_{600}$ of approximately 0.25, and both grew well in medium containing

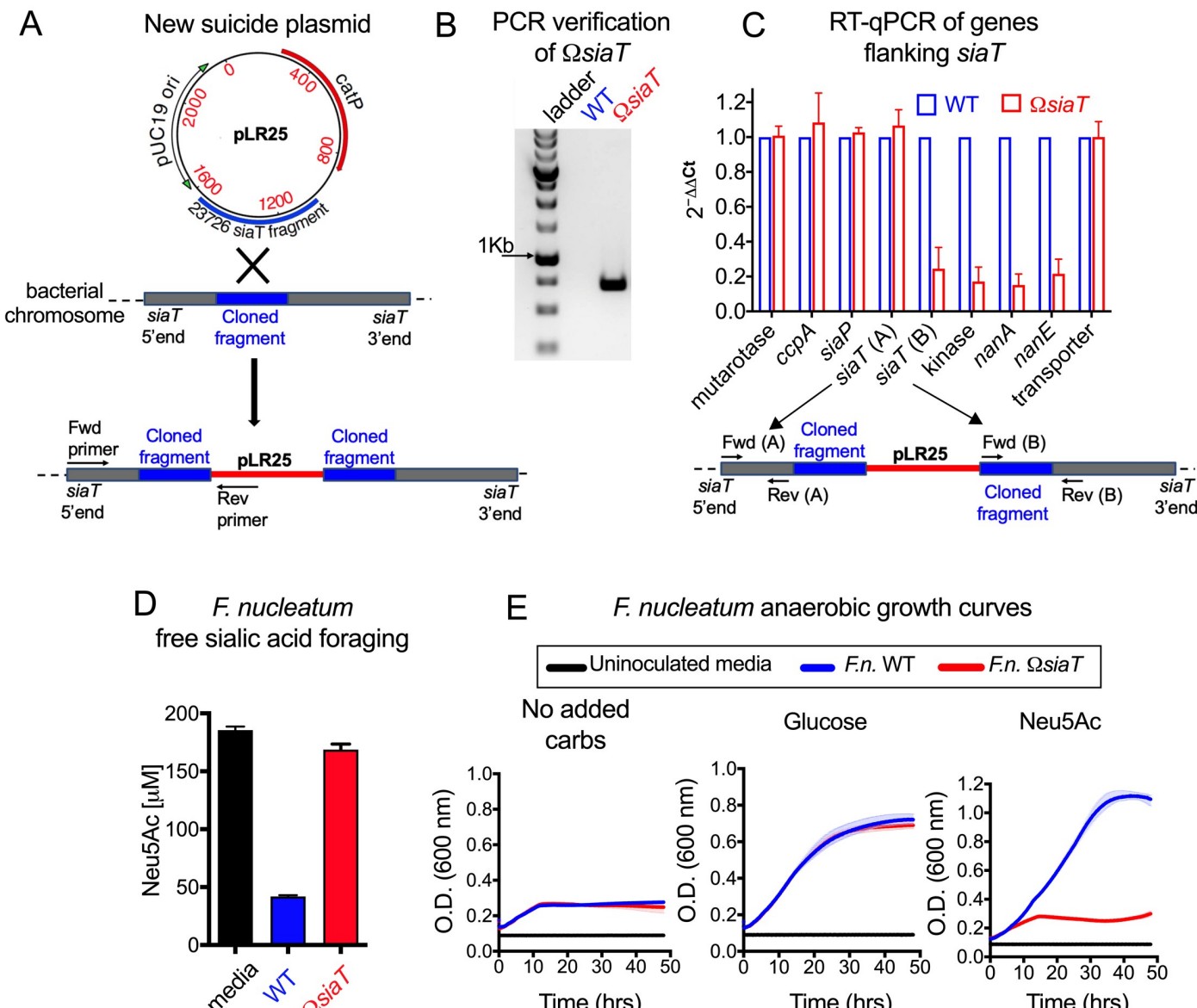

**Fig 2. Insertional mutagenesis to disrupt sialic acid catabolic pathway in *F.n.* ATCC23726 and growth analysis in nutrient-limited media.** (A) Schematic showing integration of plasmid containing *siaT* insert (pLR25) into the *F.n.* chromosome. Positions of Fwd and Rev primers used for confirming integration of plasmid are also indicated. (B) Agarose gel image with the expected PCR product confirming integration of plasmid into the *siaT* locus. (C) RT-qPCR analysis of genes flanking *siaT* in *F.n.* shows low expression of transcripts of putative sialic acid catabolism genes downstream of the plasmid insertion site. Difference in expression of each gene between *F.n.* WT versus Ω*siaT* was analyzed by the ΔΔCt method using the 16S rRNA gene for normalization. Fwd and Rev primer binding sites are indicated by arrows in the schematic below the graph. (D) Functional assessment of the WT and Ω*siaT* strains confirms disruption of sialic acid catabolism. Shown is the concentration of sialic acid (Neu5Ac) remaining in the medium 24 hpi. Sialic acid consumption by *F.n.* WT and Ω*siaT* was studied in supplemented Columbia media with free sialic acid added. (E) Growth of *F.n.* WT and Ω*siaT* in nutrient-limited media with no added carbohydrates (carbs) or with the indicated carbohydrates added. All data shown are representative of 2 or more independent experiments. The underlying numerical data for this figure can be found in S1 Data. Error bars represent standard deviation from the mean. Description of the plasmids and cloning vectors used for construction of new suicide plasmid pLR25 can be found in S1 Table. Description of the genes in the predicted sialic acid catabolic gene cluster can be found in S2 Table. ATCC, American Type Culture Collection; *catP*, chloramphenicol acetyltransferase; Fwd, forward; *F. n.*, *F. nucleatum*; hpi, hours postinoculation; *nanA*, *N*-acetylneuraminate lyase; Neu5Ac, *N*-acetylneuraminic acid; Rev, reverse; RT-qPCR, reverse transcription–quantitative PCR; *siaT*, predicted sialic acid transporter; WT, wild type.

glucose, reaching a plateau at an $OD_{600}$ of >0.6. In contrast, in the medium containing free sialic acid as the main source of carbohydrate, Ω*siaT* showed a substantial growth defect, reaching an $OD_{600}$ of only approximately 0.25, whereas WT grew to an $OD_{600}$ of >1.0 (Fig 2E). Together, these data demonstrate the genetic basis for sialic acid catabolism in *F. nucleatum* and illustrate the benefit of sialic acid catabolism under nutrient-limited conditions.

## *F. nucleatum* has restricted access to sialoglycans

Consistent with a previous study showing that 39 cultured *F. nucleatum* isolates all lacked sialidase activity [41], our bioinformatic analysis of *F. nucleatum* proteomes revealed no putative sialidases in this taxon. In an in vitro assay, using the abundant BV bacterium and known sialidase producer *G. vaginalis* as a positive control, we found that anaerobically grown *F. nucleatum* could not cleave a fluorogenic sialic acid substrate (Fig 3A). Along with previous findings, these results establish that *F. nucleatum* lacks its own sialidase activity. We therefore hypothesized that its ability to access and utilize sialic acid bound to host glycans for growth would require exogenous sialidase provided by bacteria living within the same niche (Fig 3B). To test this, *F. nucleatum* strains were inoculated in nutrient-limited media with added 3′-sialyllactose (3SL, a trisaccharide glycan containing α2–3–linked Neu5Ac) in the presence or absence of an exogenous purified sialidase from *Arthrobacter ureafaciens*. Here, we observed that both *F. nucleatum* WT and Ω*siaT* showed little growth in sialoglycan-rich media without exogenous sialidase. In contrast, upon the addition of sialidase, the WT strain grew more substantially (reaching an $OD_{600}$ >0.8) in sialoglycan-containing medium. The Ω*siaT* mutant, unable to consume sialic acids, continued to exhibit limited growth even in the presence of sialidase (only reaching an $OD_{600}$ = approximately 0.4) (Fig 3C). As described in the Methods, derivatization without prior acid hydrolysis measures free Neu5Ac, while derivatization following acid hydrolysis, which releases bound Neu5Ac from sialoglycans, measures the total Neu5Ac (Bound Neu5Ac = Total Neu5Ac − Free Neu5Ac). Measurements of the media at the endpoint showed that sialic acids remained in the bound form in the absence of exogenous sialidase, precluding *F. nucleatum* from capitalizing on this resource (Fig 3D). With the addition of sialidase, sialic acids were liberated into the free form, but only WT *F. nucleatum* was able to deplete the liberated resource. In contrast, Ω*siaT* was unable to deplete sialic acids and did not receive a growth benefit. While the sialoglycan substrate used here contains α2–3–linked sialic acids, *Arthrobacter* and *Gardnerella* sialidases are able to access both α2–3– and α2–6–linked sialic acids [34,55]. Together, these data demonstrate that *F. nucleatum* can access and consume sialic acids from sialo-glycoconjugates, only if sialidase activity from an exogenous source is available to first liberate sialic acids into the free form.

## Sialic acid catabolism prolongs vaginal colonization by *F. nucleatum*

The above findings supported our hypothesis that sialic acid catabolism may facilitate *F. nucleatum* colonization and persistence within sialidase-producing microbial communities. Although the vaginal microbiomes of women and laboratory mice are different, our previous measurements of sialidase activity in mouse vaginal washes suggested that some mice, like some women, harbor a sialidase-positive vaginal microbiome [31]. However, only approximately 10% of animals in this prior study had sialidase activity in vaginal washes. To identify a vendor whose C57BL/6 mice had a more uniform sialidase-positive phenotype, vaginal washes from animals purchased from multiple vendors were tested for sialidase activity. Endogenous vaginal sialidase activity was present in most mice from Envigo but was rarely detectable in mice from 3 other vendors (S3A Fig). Furthermore, Envigo mice had even more vaginal sialidase activity after treatment with β-estradiol, whereas mice from Jackson were sialidase-

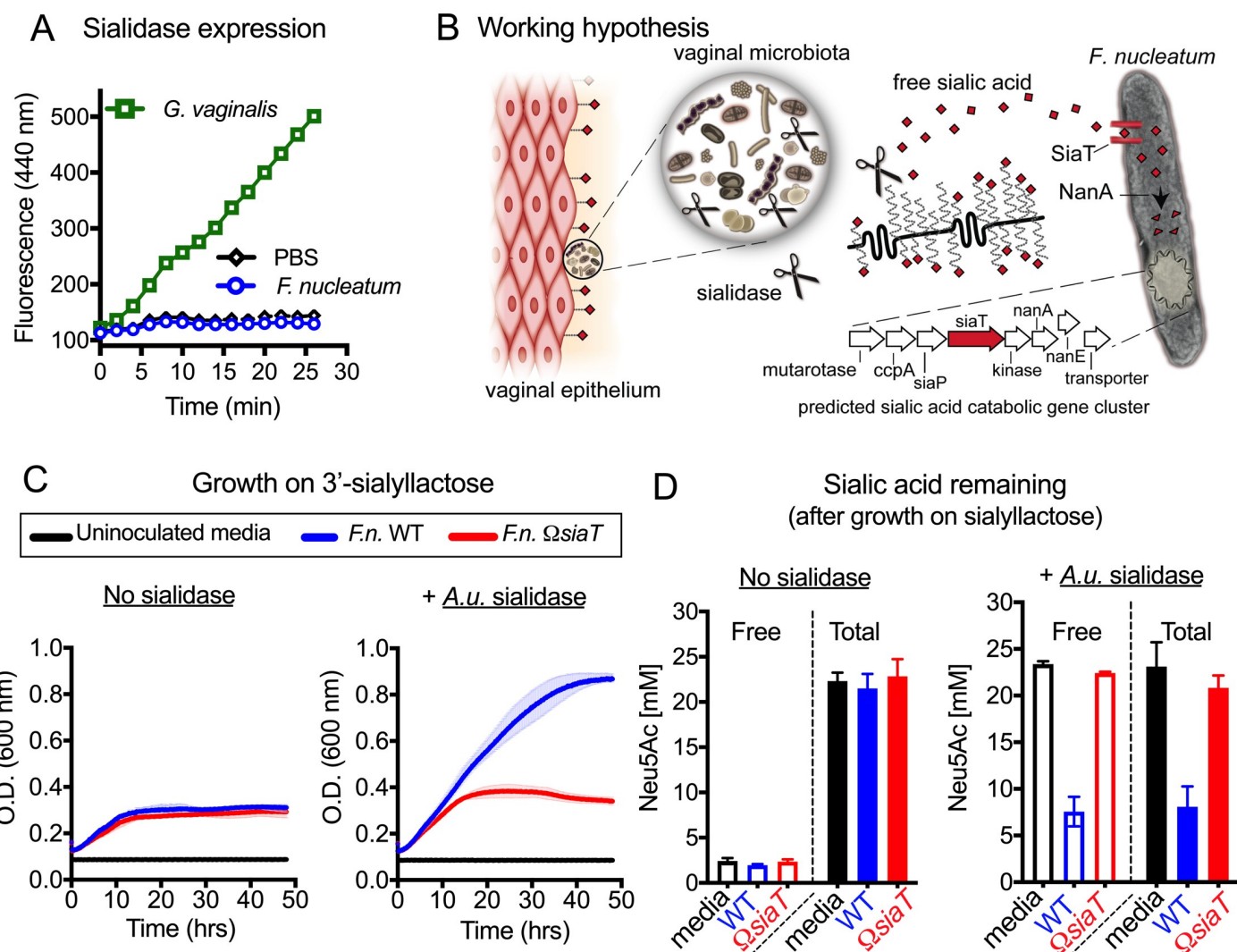

**Fig 3. *F.n.* accesses bound sialic acid from glycan chains, but only when liberated by exogenous sialidases.** (A) Analysis of cell-associated sialidase activity of anaerobically cultured *G. vaginalis* JCP8151B and *F.n.* ATCC23726 strains using fluorogenic 4MU-Neu5Ac substrate. (B) Working hypothesis: sialidase producers in the vaginal microbial community release free sialic acids (red diamonds) from host glycoconjugates, which may be accessed by *F.n.*, which does not produce sialidase. More information about *F.n.* genes shown in the predicted sialic acid catabolic gene cluster and the enzymes they encode can be found in S2 Table. (C) Growth of *F. n.* ATCC23726 WT and Ω*siaT* in nutrient-limited media with added 3SL, in the absence of sialidase (no sialidase) or presence of exogenous *A.u.* sialidase. (D) Concentrations of remaining free and total sialic acid (Neu5Ac) in nutrient-limited media (with added 3SL, from experiment shown in C) that was either uninoculated or inoculated with *F.n.* WT or Ω*siaT* strain (in absence and presence of *A.u.* sialidase). Total and free sialic acid content was measured at approximately 48 h postinoculation. Bound sialic acids are inaccessible to *F.n.* except in the presence of exogenous sialidase. All data shown are representative of 3 independent experiments. The underlying numerical data for this figure can be found in S1 Data. Error bars represent standard deviation from the mean. ATCC, American Type Culture Collection; *A.u.*, *A. ureafaciens*; *F.n.*, *F. nucleatum*; NanA, *N*-acetylneuraminate lyase; Neu5Ac, *N*-acetylneuraminic acid; OD, optical density; PBS, phosphate-buffered saline; *siaT*, predicted sialic acid transporter; WT, wild type; 3SL, 3′-sialyllactose; 4MU, 4-methylumbelliferone.

negative with or without β-estradiol treatment (S3B Fig). This is relevant because our mouse model employs β-estradiol treatment, like most bacterial vaginal inoculation models [31,56–59].

To investigate whether sialidase activity in vaginal washes was of bacterial origin, we profiled the composition of the vaginal microbiotas of Jackson and Envigo mice. To increase the available biomass, we pooled vaginal washes from cohoused mice (4 pools/vendor), and a portion of each vaginal wash pool was cultured anaerobically, then frozen in cryoprotectant (S1

Schematic). Corroborating our findings in individual vaginal washes, microbiota pools (cultured vaginal wash) from Envigo mice, but not Jackson mice, had high sialidase activity (S3C Fig). The presence of sialidase activity in mixed cultures from Envigo mice is consistent with the hypothesis that sialidase is microbial and not host-derived. To further validate this hypothesis, we analyzed the mouse vaginal microbiota by sequencing the V1–V2 region of the gene encoding the 16S ribosomal subunit (S3D Fig) (see Methods for V1–V2 rationale). DNA was extracted from both the original (uncultured) pooled wash material and from viable communities of bacteria (microbiota pools) derived from the cultured vaginal wash material (S1 Schematic). In addition to the 16S sequencing analysis, the cultured microbiota pools were also plated on agar media, and the resulting colonies were picked into 96-well plates to perform sialidase activity assays. PCR and sequencing were also performed to identify colonies. Sequencing revealed that several of the uncultured vaginal wash pools and ex vivo-cultured communities from Envigo mice contained *Enterococcus casseliflavus/gallinarum*, whereas the corresponding samples from Jackson mice did not. Notably, colonies of *E. gallinarum* and *Bacteroides* spp. from cultured vaginal wash pools from Envigo mice expressed sialidase activity (S3E Fig and S3F Fig). Furthermore, we previously isolated sialidase-positive *E. gallinarum* from mouse vaginal washes that contained sialidase activity [20,31]. These findings are consistent with differences in vaginal sialidase activity between mice from the different vendors and confirm that C57BL/6 mice from Envigo exhibit vaginal sialidase activity arising from the endogenous microbiota. Finally, we note that estrogenized mice from Envigo were consistently sialidase-positive in 4 independent studies of mice purchased over 2 years and were therefore a relevant model to test our hypothesis.

To determine whether the ability to catabolize sialic acid contributes to *F. nucleatum* colonization or persistence within sialidase-producing microbial communities, Envigo mice were vaginally inoculated with *F. nucleatum* WT or Ω*siaT*, and vaginal washes were collected longitudinally for several weeks (Fig 4A). Before inoculation, both the infection groups had similar amounts of sialidase activity (Fig 4B). As early as 72 h postinoculation (hpi), and later at 6 days postinoculation (dpi) up to 8 dpi, Ω*siaT* had significantly lower titers than WT in Envigo mice (Fig 4C and S4 Fig). Sufficient vaginal wash material was available for sialic acid measurement at 8 dpi from some of the mice in each experimental group. Free sialic acid levels were similar between WT- and Ω*siaT*-inoculated mice. However, total sialic levels were lower in mice colonized with WT- compared to Ω*siaT*-inoculated mice (S4 Fig). The latter comparison was underpowered with the number of samples available and did not reach statistical significance ($P = 0.19$). However, the lower levels of total sialic acid in mice with *F. nucleatum* WT are consistent with a depletion of sialic acid, a phenotype also observed in women with BV. Furthermore, Ω*siaT* exhibited a significant defect in vaginal persistence relative to the WT strain ($P = 0.02$; Fig 4D). Similar to Envigo mice, *F. nucleatum* was also able to colonize sialidase-negative mice from Jackson Labs. However, there was no detectable benefit afforded by sialic acid catabolism in this setting, either in overall bacterial burden or persistence (S5 Fig). We conclude from these experiments that the ability to catabolize sialic acid benefits *F. nucleatum* colonization and persistence in a sialidase-positive vaginal environment.

## *F. nucleatum* enhances sialidase activity in mouse vaginal communities

While working with the mouse model, we observed that vaginal sialidase activity waned rapidly after arrival in our facility, both in mice that received mock inoculations (Fig 5A) and those inoculated with a different urogenital pathogen such as *E. coli* (S6 Fig). In contrast, when inoculated with *F. nucleatum*, vaginal sialidase activity was maintained for a significantly longer duration (Fig 5A). Additionally, inoculation of WT *F. nucleatum* helped to maintain higher

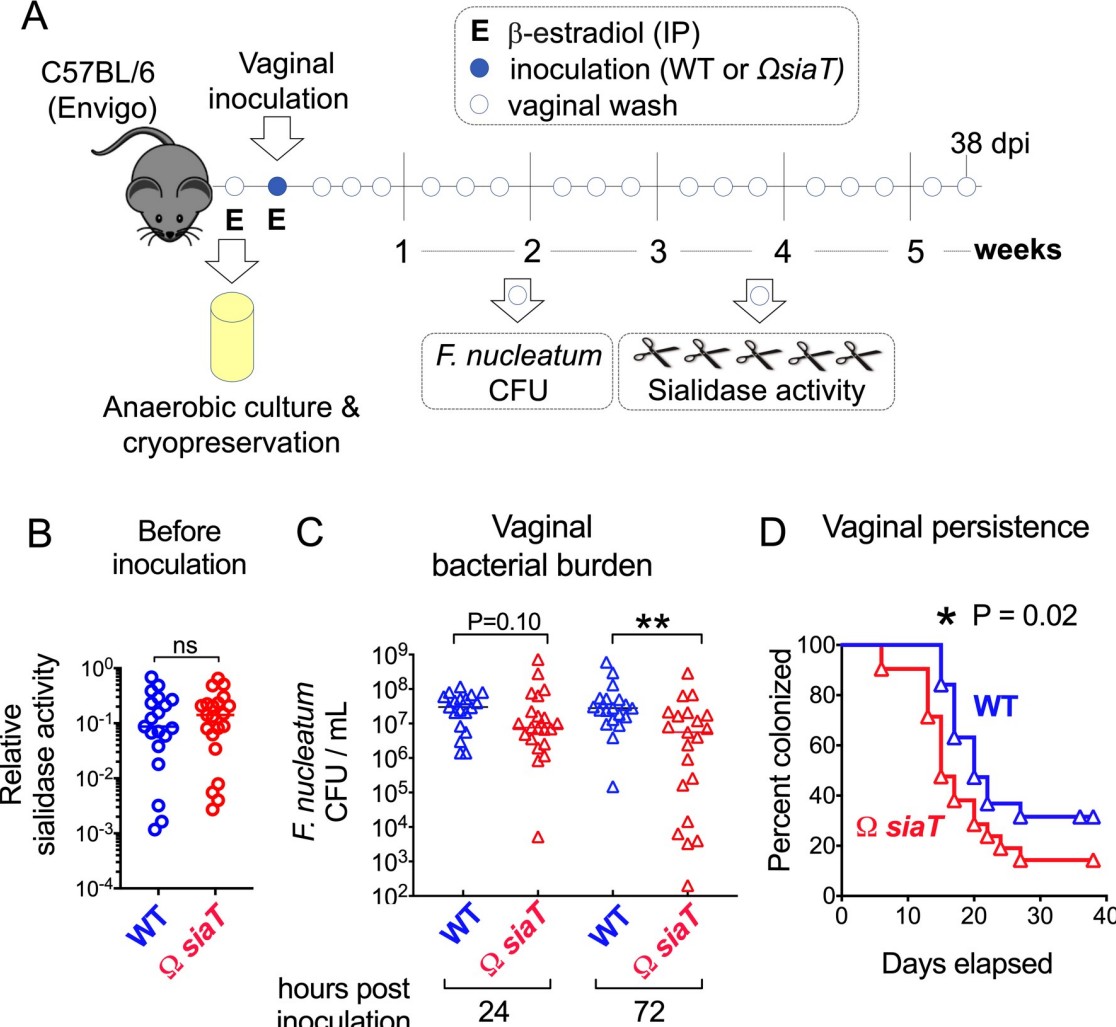

**Fig 4. Sialic acid catabolism encourages *F. nucleatum* colonization in mice with sialidase-positive vaginal microbiomes.** (A) Time course of vaginal colonization with *F. nucleatum* ATCC23726. Open circles indicate vaginal wash collection which were used to monitor colonization status and sialidase activity at all time points. (B) Analysis of sialidase activity in vaginal washes of Envigo mice before inoculation with *F. nucleatum* ($P = 0.73$, Mann–Whitney U test). (C) CFU enumeration of *F. nucleatum* WT and Ω*siaT* in vaginal washes collected at the indicated time points from Envigo mice. **$P < 0.01$, Mann–Whitney. (D) Time course of vaginal colonization with *F. nucleatum* WT and Ω*siaT* mutant in Envigo mice. Percent of mice colonized (*y* axis) was monitored on day 1 and every 2 days thereafter up to 38 dpi (*x* axis). For Kaplan–Meier analysis, mice were considered cleared when no CFUs were detected in undiluted wash at 2 consecutive time points. Statistical significance was assessed by Gehan–Breslow–Wilcoxon test, *$P < 0.05$. The graphs represent combined data from 2 independent experiments, with 10 mice per group in each experiment. The underlying numerical data for this figure can be found in S1 Data. ATCC, American Type Culture Collection; CFU, colony-forming unit; dpi, days postinoculation; E, estrogenization; IP, intraperitoneal; *siaT*, predicted sialic acid transporter; WT, wild type.

levels of vaginal sialidase for a longer duration as compared to inoculation with Ω*siaT* (Fig 5B). While sialidase activity was equivalent between WT- and Ω*siaT*-inoculated mice at 1 dpi (Fig 5C), by 6 dpi sialidase activity was significantly lower in vaginal washes from mice inoculated with Ω*siaT* compared to WT. These data suggest that besides receiving benefit from the sialidase-producing microbiota, *F. nucleatum* also exerts influence of its own by supporting the sialidase-positive vaginal microbiota (Fig 5D).

To investigate the effect of *F. nucleatum* on sialidase activity produced by vaginal communities in a more reduced system, we investigated the responses of mouse vaginal microbiotas to

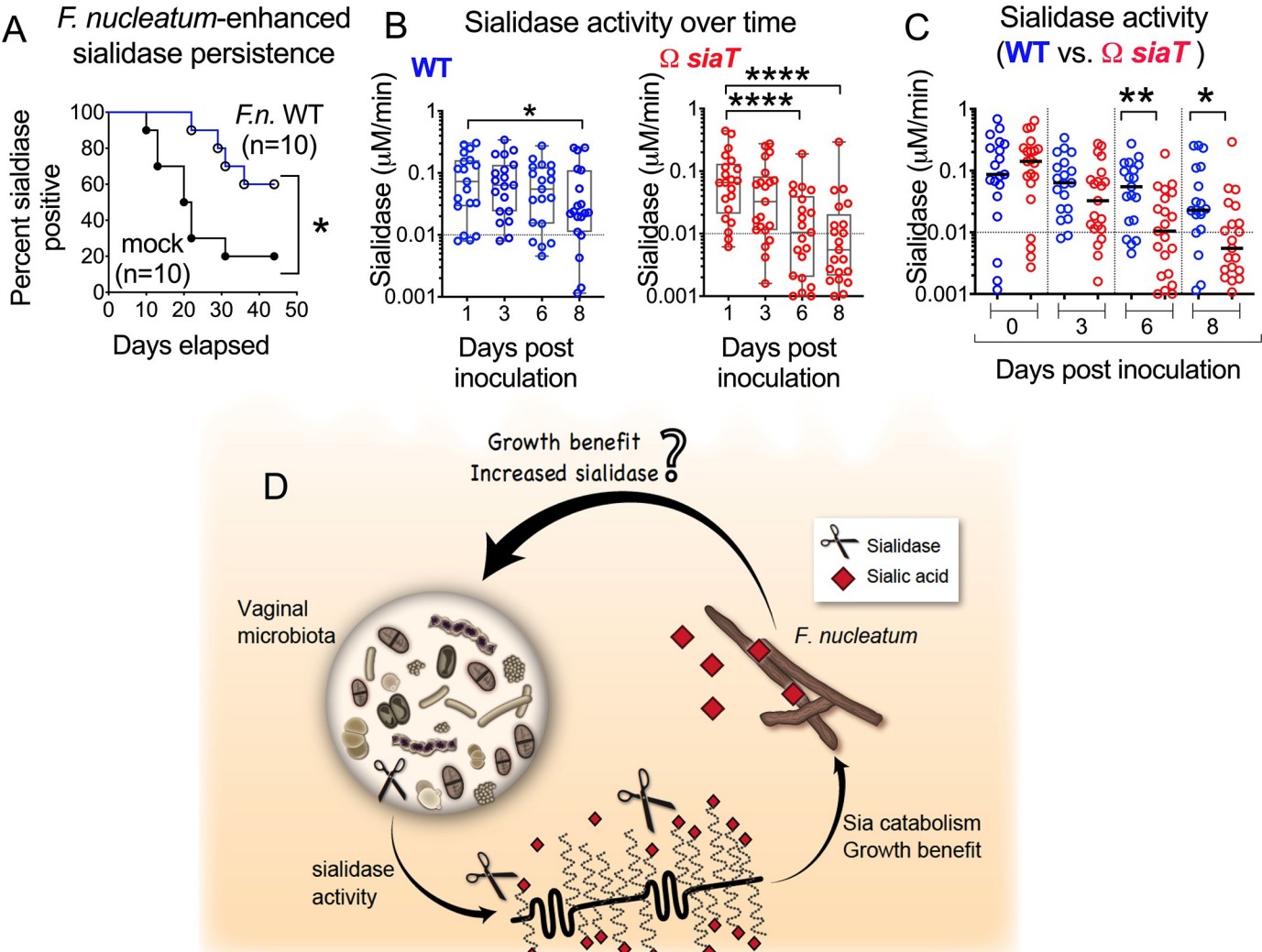

**Fig 5. Microbiota-derived sialidase activity in the mouse vagina is sustained by *F.n.* colonization.** (A) Persistence of sialidase activity in C57BL/6 mice (Envigo) inoculated either with WT *F.n.* or vehicle only. Mice were considered sialidase-negative if sialidase activity levels in vaginal washes were below 0.01 μM/min (using the 4MU-Sia assay) for 3 sequential time points. Data are shown from a single experiment with a total of *N* = 20 animals. *P < 0.02, log–rank test. (B–C) Sialidase activity in vaginal washes from 1 to 8 dpi from individual animals purchased from Envigo, estrogenized, and inoculated with either WT or Ω*siaT F.n.* The graphs represent combined data from 2 independent experiments, with 10 mice per group in each experiment. Data points with negative values were set to 0.001 to represent them on the log scale. (B) Sialidase activity at later time points were compared to day 1 values using the Friedman test, with correction for multiple planned comparisons using Dunn's test. (C) Same experiment and data as shown in B but analyzed to compare between WT- or Ω*siaT*-inoculated animals at each time point using the Mann–Whitney test. On all graphs, *P < 0.05, **P < 0.01, ****P < 0.0001. (D) Sialidase producers in the vaginal microbial community release free sialic acids (red diamonds) from host glycoconjugates, providing benefits to *F.n.* Hypothesis: at the same time, vaginal community members themselves derive benefits from *F.n.*, leading to sustained sialidase activity in these communities. Sialidases are represented as scissors. The underlying numerical data for this figure can be found in S1 Data. dpi, days postinoculation; *F.n.*, *F. nucleatum*; *siaT*, predicted sialic acid transporter; WT, wild type; 4MU, 4-methylumbelliferone.

*F. nucleatum* (WT or Ω*siaT*) in an ex vivo anaerobic culture model. As described in the Methods (Fig 6A and S1 Schematic), microbial communities generated from vaginal washes of mice were cultured anaerobically in the presence or absence of equal amounts of *F. nucleatum* WT or Ω*siaT*, and sialidase activity was measured after overnight culture (approximately 16 h). In cultures from all 4 microbiota pools, those receiving *F. nucleatum* (either WT or Ω*siaT*) had significantly higher levels of sialidase activity than the identical microbiota cultured without *F. nucleatum* (Fig 6B). This assay was performed under nutrient replete conditions in supplemented Columbia media in which WT and Ω*siaT* strains grew equally well, resulting in

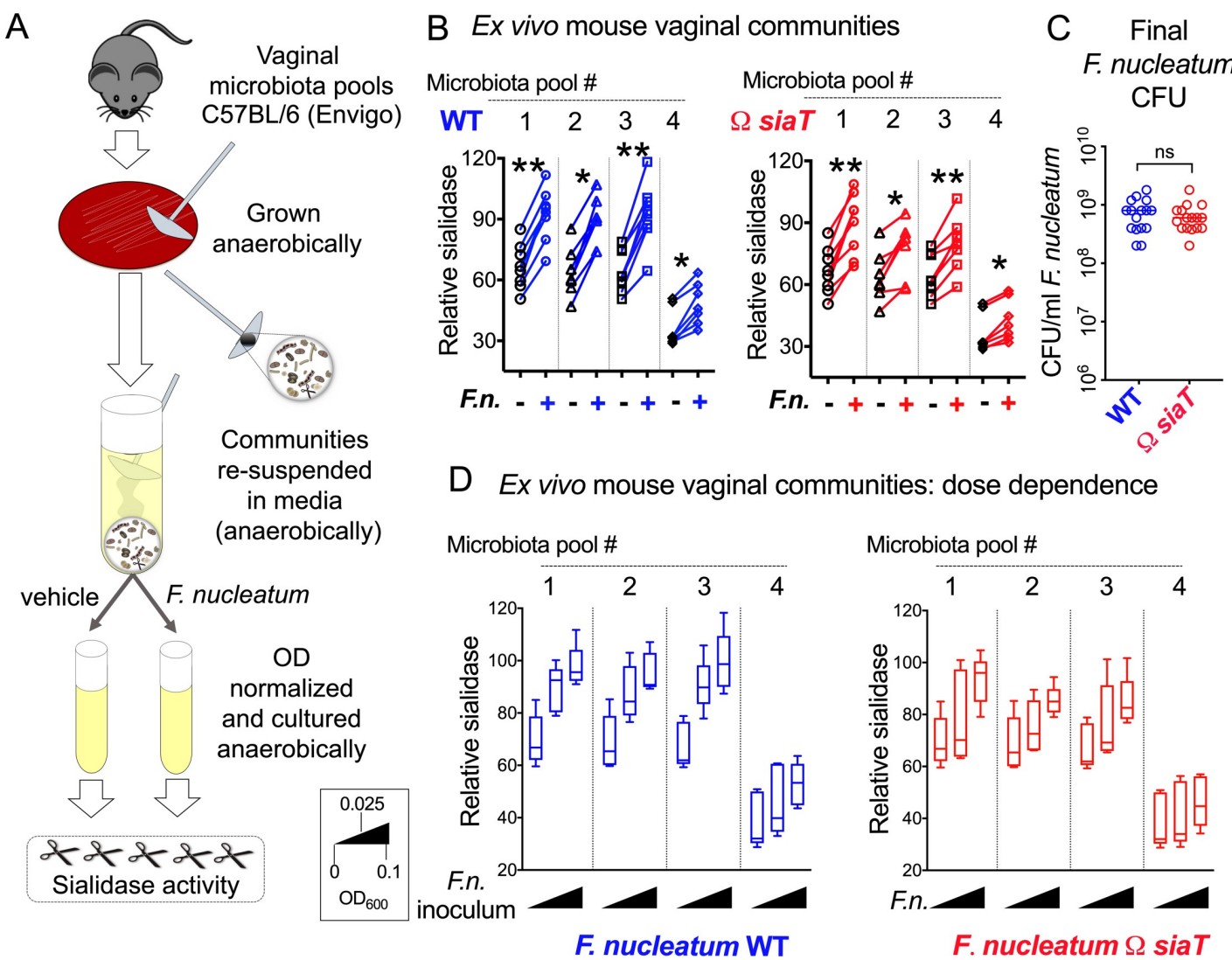

**Fig 6. Ex vivo interaction between *F.n.* and the mouse vaginal microbiota leads to enhanced sialidase activity.** (A) Microbiotas from vaginal washes were collected from mice prior to inoculation in Fig 4B, pooled (by cage), cultured, and frozen (S1 Schematic, Step 1). Bacteria were recovered from frozen microbiota pools by streaking them on supplemented Columbia blood plates anaerobically and incubating for 24 h at 37˚C. Microbial growth was gently scraped from these plates and resuspended in liquid media to prepare OD-normalized inocula and cocultured with *F.n.* WT or Ω*siaT* overnight (approximately 16 h). (B) Sialidase activity in microbiota pools from Envigo mice, cultured in the presence or absence of *F.n.* Each "microbiota pool" consists of a cultured vaginal community from pooled vaginal wash of 4–5 cohoused mice. Wilcoxon paired-sign rank test was used for pairwise comparison. Data shown are combined from 4 independent biological replicates with 7–8 technical replicates for each microbiota pool. *P < 0.05, **P < 0.01. (C) *F.n.* titers after overnight growth with microbiota pools from Envigo mice. (D) Dose-dependent boost in sialidase activity in microbiota pools from Envigo mice, cultured in the presence or absence of *F.n.* (inocula high to low: $OD_{600}$ 0.10 or 0.025 or no *F.n.*). Line in the box indicates median value. Data shown are representative of 3 independent experiments. The underlying numerical data for this figure can be found in S1 Data. CFU, colony-forming unit; *F.n.*, *F. nucleatum*; OD, optical density; *siaT*, predicted sialic acid transporter; WT, wild type.

indistinguishable final colony-forming unit (CFU) titers (Fig 6C). Together with the result in Fig 6B, this suggests that WT and Ω*siaT*, when present at the same levels, result in similar increases in microbiota-derived sialidase activity. Variation of the starting inoculum of both *F. nucleatum* strains revealed a dose-dependent effect on microbiota-derived sialidase activity (Fig 6D). These data suggest that in mice, the divergence of sialidase activity in groups receiving WT versus Ω*siaT* arising after 6 days (Fig 5C) resulted from different "doses" of *F.*

*nucleatum* (arising as of day 3; Fig 4C and S4 Fig) rather than a specific feedback loop involving sialic acid catabolism.

## *F. nucleatum* enhances sialidase activity and facilitates growth of *G. vaginalis* in human vaginal communities

A significant limitation of using mice to model interactions within the vaginal microbiome is that they have distinctly different vaginal microbiotas compared to humans. Thus, our findings in mice led us to wonder whether *F. nucleatum* also leads to increased sialidase activity in human vaginal microbial communities. To test whether *F. nucleatum* stimulates sialidase activity in human vaginal microbial communities, vaginal swabs were collected from women, and living bacterial communities were then transported anaerobically to the lab and immediately prepared for viable cryopreservation under anaerobic conditions (S2 Schematic: Step 1). Next, vaginal communities from 21 women who had sialidase activity in their aerobic vaginal swab eluates (S7 Fig) were thawed and cultured anaerobically in the presence or absence of *F. nucleatum* (Fig 7A and S2 Schematic: Step 2). As we observed with mouse microbial communities, human vaginal communities cultured with added *F. nucleatum* had significantly higher levels of sialidase activity after culture than the identical community cultured without *F. nucleatum* (Fig 7B).

There are at least 2 possible explanations for the enhanced sialidase activity observed upon addition of *F. nucleatum* to vaginal communities. First, if *F. nucleatum* drives sialidase activity in communities by promoting the growth of sialidase-producing bacteria, it would indicate a mutualistic relationship between *F. nucleatum* and sialidase-producing bacteria. Alternatively, *F. nucleatum* could induce sialidase expression without promoting growth of sialidase-producing bacteria, which would suggest a commensal or parasitic relationship between *F. nucleatum* and sialidase-producing bacteria (depending on whether the latter were unaffected or harmed). *G. vaginalis* is believed to be a primary source of sialidase activity in vaginal fluids of women with BV [31,60]. To further investigate the potential benefit of *F. nucleatum* to sialidase-producing bacteria, we used community profiling to assess the relative levels and qPCR for absolute quantification of *G. vaginalis* in human vaginal communities. Following DNA extraction from vaginal communities cultured with or without *F. nucleatum*, we performed community profiling based on the gene encoding the 16S ribosomal subunit (V4 region). When *F. nucleatum* was present, *G. vaginalis* occupied a significantly larger fraction of the endogenous community compared to identical microbiotas cultured under the same conditions but without *F. nucleatum* (Fig 7C). Next, we used a previously validated qPCR assay based on the *Gardnerella* Translation elongation factor Tu (*tuf*) gene encoding elongation factor Tu [61], *G. vaginalis* was detected at significantly higher levels in communities cultured with *F. nucleatum* compared with the same communities cultured without addition of *F. nucleatum* (Fig 7D). These data strongly suggest that *F. nucleatum* supports the development of dysbiotic vaginal communities that become enriched with *Gardnerella* and sialidase activity.

To more specifically study the effect of *F. nucleatum* on *G. vaginalis*, we established an in vitro coculture system. *G. vaginalis* (strain JCP8151B) and *F. nucleatum* were coinoculated across a range of inocula doses into supplemented Columbia broth, which is nutrient-rich but inadequate for growth of *G. vaginalis*. Cultures in 96-well plates were incubated anaerobically at 37°C overnight. Inoculated alone under these conditions, *G. vaginalis* could not be recovered alive (Fig 8A, plotted at one-half the detection limit). However, coinoculation with *F. nucleatum* resulted in significant growth of *G. vaginalis* (>4 orders of magnitude), which increased further with higher inocula of *F. nucleatum*. Experiments using filter-sterilized (cell-free) spent media from cultures of *F. nucleatum* revealed a similar boost in *G. vaginalis* CFU,

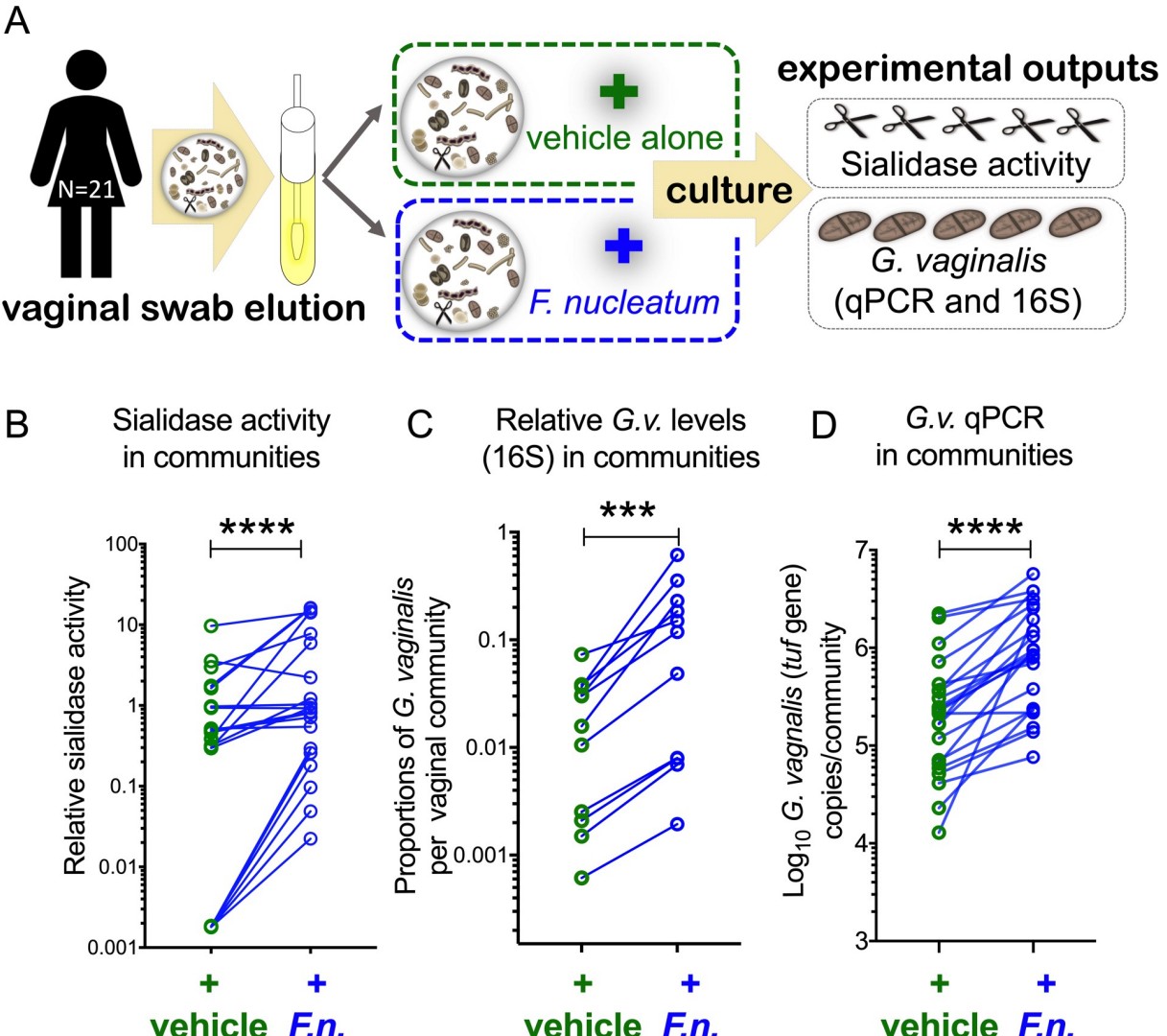

**Fig 7. *F.n.* supports *Gardnerella* growth and stimulates sialidase activity within human vaginal communities.** (A–D) Human vaginal communities from 21 individual women were used. (A) Microbial communities were eluted from vaginal swabs under anaerobic conditions in nutrient-rich media. Communities with potential sialidase producers (sialidases are represented as scissors) were cultivated anaerobically in supplemented Columbia media in the presence or absence of added *F.n.* (B) Sialidase activity was measured following anaerobic culture. Negative values were set to 0.0018 (lowest positive value) to depict them on the log scale. Graphs show data combined from 2 independent experiments. (C) Relative abundance of *G.v.* in cultured human vaginal communities. Sequencing of the gene encoding the V4 region of 16S rRNA was used to estimate proportions of *G.v.* in the microbial communities. (D) Quantitation of *G.v.* in cultured human vaginal communities by *tuf* qPCR. In each case, statistical comparison between the 2 groups was performed using Wilcoxon matched-pairs signed rank test. The underlying numerical data for this figure can be found in S1 Data. On all graphs, \*\*\**P* < 0.001, \*\*\*\**P* < 0.0001. *F.n.*, *F. nucleatum*; *G.v.*, *G. vaginalis*; qPCR, quantitative PCR; *tuf*, Translation elongation factor Tu.

suggesting that the *G. vaginalis* growth-promoting factor or factors are present in the cell-free supernatant and do not require live bacteria (Fig 8B). There was a dose-dependent increase in *G. vaginalis* titers with increasing proportions of *F. nucleatum* spent supernatant. These data suggest that *F. nucleatum* may secrete one or more factors that have a strong growth-promoting effect on *G. vaginalis*.

Viable recovery of *G. vaginalis* in monoculture could also be improved by increasing the dose of the *G. vaginalis* inoculum. However, across the entire range of *G. vaginalis* inocula used, the addition of *F. nucleatum* improved recovery by orders of magnitude (as few as 1, as

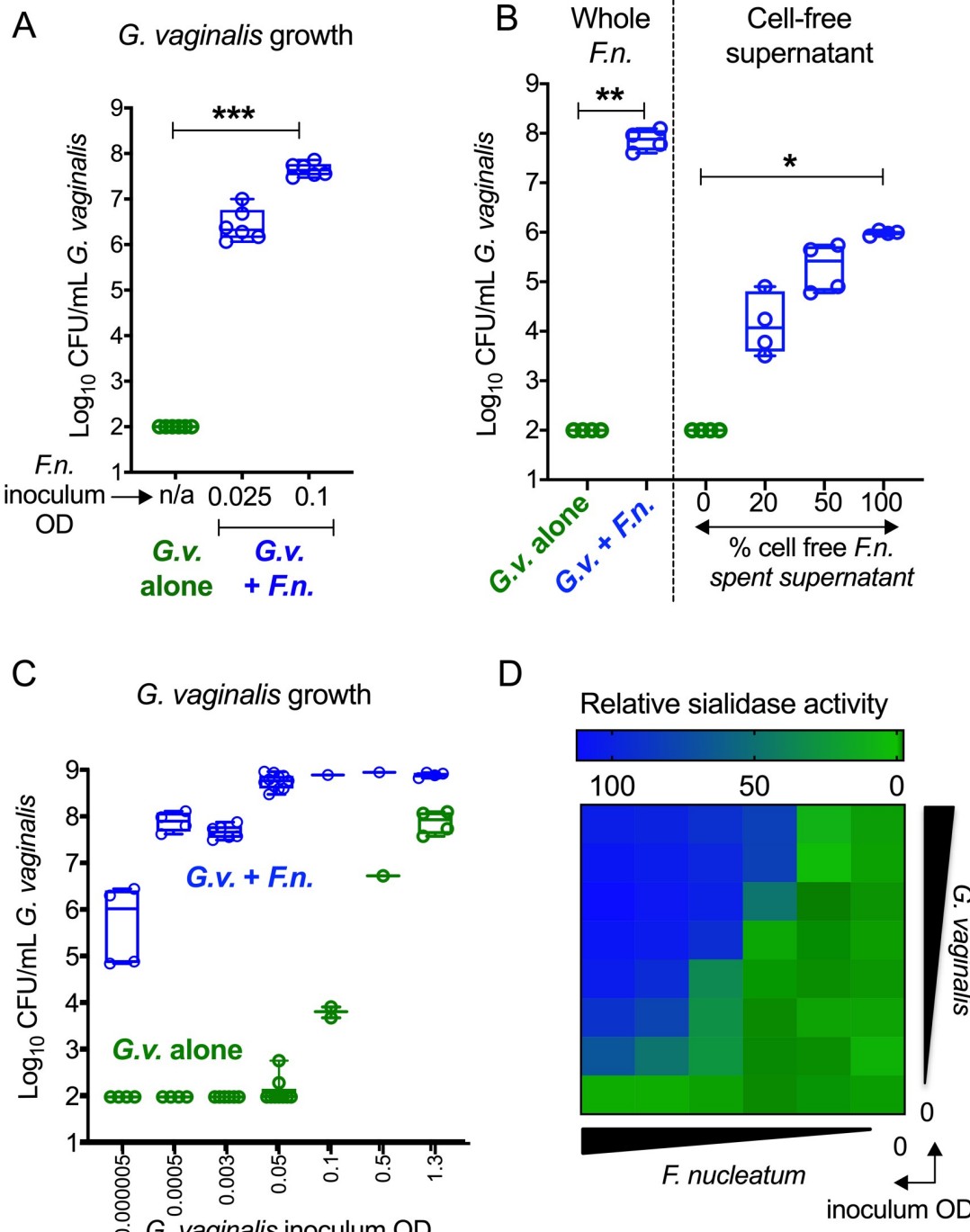

**Fig 8. Enhanced growth of *G.v.* in cocultures with *F.n.*** (A) Analysis of *G.v.* growth by CFU enumeration in anaerobic cocultures with or without *F.n.* in supplemented Columbia medium. Data shown are combined from 2 independent experiments, with 3 technical replicates each. (B) Growth of *G.v.* in media with whole *F.n.* or spent cell-free supernatant from *F. n.* culture. Data shown are combined from 2 independent experiments, with 2 technical replicates each. (C) Dose-dependent effect of *F.n.* on *G.v.* growth in supplemented Columbia medium. Data shown are combined from 9 independent experiments. Note that at low inocula, *G.v.* was not detectable under these conditions in the absence of *F.n.* In this case, *G.v.* levels were plotted at one-half the LOD (LOD = 200 CFU/mL). For A–C, data are represented in box plot format with whiskers of min and max. Statistical analysis: Kruskal–Wallis followed by Dunn's multiple comparisons test. (D) Sialidase activity in *G.v.* cultured overnight with *F.n.* WT in supplemented Columbia medium. Heat map data are representative of 2 independent experiments. Inoculum range—*F.n.* $OD_{600}$: 0, 0.00625, 0.0125, 0.025, 0.05, 0.1; *G.v.* $OD_{600}$: 0, 0.000781, 0.001562, 0.003125, 0.00625, 0.0125, 0.025, 0.05. The underlying numerical data for this figure can be found in S1 Data. On all graphs, $^*P < 0.05$, $^{**}P < 0.01$,

***$P < 0.001$. CFU, colony-forming unit; *F.n.*, *F. nucleatum*; *G.v.*, *G. vaginalis*; LOD, limit of detection; n/a, not applicable; OD, optical density; WT, wild type.

many as 7; Fig 8C). *G. vaginalis* viability after overnight incubation in this system required at least a 20,000-fold higher inoculum in monoculture compared to what could be achieved in cocultures with *F. nucleatum*. Comparing sialidase activity across a spectrum of *G. vaginalis* inocula doses (up to $OD_{600}$ 0.05) revealed no detectable sialidase activity when the organism was grown without *F. nucleatum* (Fig 8D). Even a relatively small inoculum of *F. nucleatum* resulted in measurable sialidase activity in the cultures. Also, sialidase activity could be rescued in cultures inoculated with progressively lower levels of *G. vaginalis* by adding higher amounts of *F. nucleatum*. As with WT *F. nucleatum* (Fig 8), dose-dependent increases in sialidase activity were also observed when *G. vaginalis* was cocultured with Ω*siaT F. nucleatum* (S8 Fig). This is similar to observations made in coculture experiments with mouse vaginal communities in which both WT and Ω*siaT* had similar boosting effects on sialidase activity (Fig 6B). These findings further illustrate that the growth benefit to *Gardnerella* is not directly tied to *F. nucleatum* sialic acid catabolism. This also suggests that while both *F. nucleatum* and *G. vaginalis* may benefit from cohabitation, their mutualism is not obligate, but rather facultative. However, in a polymicrobial community with shared and limited nutrient resources, such a relationship could become a necessary tool for survival and persistence.

## Discussion

A growing number of literature reports describe the contents and dynamics of the human vaginal microbiome, but we still know surprisingly little about how vaginal dysbiosis develops or why women with BV are more likely to harbor potentially pathogenic vaginal bacteria. One of the prevailing concepts is that an absence of secreted antimicrobial activities produced by "healthy" lactobacilli creates a permissive vaginal environment in which opportunists can overgrow [7,14,62]. Here, we provide evidence for a different type of relationship—a mutualism that actively reinforces new pathogen colonization (*F. nucleatum*) while promoting the expansion of features linked with dysbiosis (*G. vaginalis*, sialidase). Our analysis of sialic acid catabolism by *F. nucleatum* suggests that the presence of sialidase activity in the vagina during BV may allow *F. nucleatum* to capitalize on the increased availability of free sialic acid. Additionally, experiments in the mouse model further support the role of sialic acid metabolism in the persistence of *F. nucleatum* in a sialidase-positive environment in vivo. We also discovered that a mutually beneficial relationship exists between *F. nucleatum* and sialidase-producing bacteria within vaginal microbiomes. Both ex vivo and in vitro coculture studies show that *F. nucleatum* can support the outgrowth of *G. vaginalis*, one of the most common and abundant species that overgrows in BV, suggesting an important role of interspecies crosstalk in shaping the dynamics of the vaginal microbiome.

Several features of the mouse model, together with considerations of human physiology, have guided our interpretations of the in vivo data. Sialic acid catabolism provided a benefit to *F. nucleatum* in specific contexts. However, the organism initially colonized 100% of mice, even in the absence of sialidase, suggesting that sialic acid catabolism is not an absolute requirement for initial *F. nucleatum* vaginal colonization in mice. Thus, *F. nucleatum* must be using other strategies to colonize the vagina in the sialidase-negative setting. Indeed, the mechanisms of colonization may be distinct in mice from different vendors, in which competition for resources with different microbes may have distinctly different outcomes. Other models of vaginal infection have also yield different results depending on the composition of the endogenous microbiota, as illustrated in a recent study of Group B *Streptococcus* [63]. Furthermore, because of the diverse potential sources of bacterial sialidase, the compositional and physiological

complexity of vaginal communities, and insufficient experimental tools for genetic manipulations in many fastidious vaginal anaerobes, we lack precise experimental control of sialidase within bacterial communities. In addition to mutualistic relationships among members of the dysbiotic microbiota, antagonistic interactions probably also impact the vaginal environment in humans. For example, women without BV, who lack sialidase activity, also have lactobacilli that create an environment of high lactic acid and low pH [20,64], which likely contributes to the exclusion of *Fusobacterium*. Together, these considerations provide an overarching rationale that may help explain why *F. nucleatum* has been isolated at approximately 6-fold higher rates among women with BV compared to those without (54% versus 9.4%, respectively) [3]. Future studies on the ecology of vaginal microbial communities will be necessary to map and evaluate the magnitude of synergistic and antagonistic interactions connecting species within vaginal communities.

Although it is not known how women acquire vaginal *F. nucleatum*, the ubiquitous presence of *F. nucleatum* in the mouth points to orovaginal contact as one plausible route of vaginal exposure. Indeed, several studies suggest orogenital contact is a risk factor for BV or vaginal community features consistent with dysbiosis, such as low levels of *Lactobacillus* or high levels of *G. vaginalis* [65–69]. Our data suggest that such exposure could lead to outgrowth of *G. vaginalis*, even if *Fusobacteria* are introduced at relatively low numbers. In light of these findings, future studies should evaluate whether exposure of the vagina to microbes from other body sites (for example, oral, rectal, penile) may enhance dysbiotic features of the vaginal microbiome.

*F. nucleatum* often cohabitates with sialidase-producing bacteria at mucosal niches such as the oral cavity, gut, and vagina [42–46]. Thus, our findings also have pertinence for other dysbioses and infections where *Fusobacterium* species have been implicated. For example, *F. nucleatum* is a ubiquitous member of the oral microbiome in humans but also overgrows and plays key roles in the multispecies biofilm seen in periodontal disease. Some of the members of this biofilm are also sialidase producers [70–72]. As observed in BV [31], depletion of sialic acids from the oral mucosa has been observed in gingivitis [73], another condition associated with overgrowth of *F. nucleatum*. Additionally, *F. nucleatum* has also been identified as a contributor in colorectal cancer [74]. Colorectal tumors that have metastasized to other locations in the body commonly contain not only *F. nucleatum* but also species of *Bacteroides*, which are sialidase producers [75]. In fact, sialidase-producing members of the Bacteroidetes often cohabitate with *Fusobacterium* in abscesses, amniotic fluid infections, and polymicrobial bacteremia/sepsis [76–80]. Together with the data presented here, these findings suggest that *F. nucleatum* may benefit from sialidase-producing bacteria during disease processes throughout the human body.

In conclusion, these studies provide the first direct evidence that mutual benefit, made possible in part through metabolite cross-feeding, can promote pathogen colonization of the vagina and encourage features of vaginal dysbiosis. These findings may help explain why women with BV are more likely to be vaginally colonized with potential pathogens linked with intrauterine infection and adverse health outcomes. Although the present work greatly advances our understanding of bacterial relationships that may influence the development of dysbiosis, additional study is needed to apply these principles to the prevention or treatment of dysbiosis in women. Additional investigations of the relationships between vaginal microbes may further reveal the metabolic networks that connect bacterial species within diverse vaginal communities and point to potential target pathways to disrupt these relationships.

## Methods

### Ethics statement

Animal research: mouse experiments were carried out in strict accordance with the recommendations in the Guide for the Care and Use of Laboratory Animals. The protocol was

approved by the Institutional Animal Care and Use Committee (IACUC) of Washington University School of Medicine (Protocol Number: 20170081).

Human subject research: vaginal specimens collected from women who participated in the Urogenital Community (UC) Bank project, an archive of living microbial communities established for translational microbiome research, were used for these studies, according to Washington University in Saint Louis IRB-approved protocol (protocol number, 20170412; PI, Amanda Lewis). A written informed consent was obtained from all participants enrolled in the UC Bank project. Nonpregnant women seeking care at the North Central Community Health Center, a no-fee clinic operated by the St. Louis County health department, were enrolled in the UC Bank project. While no specific racial or demographic group was sought or excluded for participation, >90% of women enrolled self-identified as black, a demographic very similar to the community neighboring the clinic. Women with known human immunodeficiency virus or hepatitis C virus or with recent antibiotic use (past 3 weeks) were excluded.

### Bacterial strains and growth conditions

For a list of bacterial strains, plasmids, and primers, see S1 Table. WT strains of *E. coli* (TOP10) [Invitrogen, Carlsbad, CA, USA] and MG1655, UTI89, and CFT073 [provided by Dr. Scott Hultgren] were grown in lysogeny broth (LB) at 37˚C unless otherwise noted. For plasmid maintenance, ampicillin was used at 100 μg/mL and chloramphenicol at 20 μg/mL. Sm$^R$ isolate of *G. vaginalis* JCP8151B [31,81], hereafter referred to simply as *G. vaginalis*, was cultured in NYCIII media containing 10% heat-inactivated horse serum at 37˚C in an anaerobic chamber unless stated otherwise. *F. nucleatum* strains ATCC25586, ATCC23726, and ATCC10953 were obtained from ATCC (Manassas, VA, USA). Strains JMP2A, FQG51A, JMSY1, JMH52, SYJL4, JM6, and JM4 [50] were provided by Dr. Justin Merritt. ATCC23726 is the primary WT strain background used to model interaction with sialidases and the microbiome. As described below, a spontaneous Sm$^R$ mutant of ATCC23726 was isolated and, unless otherwise indicated, was used in all experiments. All fusobacterial strains, unless stated otherwise, were grown at 37˚C in an anaerobic chamber in Columbia broth supplemented with hemin (5 μg/mL) and vitamin K3 (1 μg/mL), referred to as "supplemented Columbia" (contains glucose as well as sialoglycans from animal ingredients). For clarity, we have avoided the term "supplemented" to refer to the addition of other components to media. When grown on solid media, supplemented Columbia blood agar plates were used (containing 5% defibrinated laked sheep blood and 1.5% agar). For some experiments, we developed a nutrient-limited media that contains base components present in Columbia media, including proteose peptone #3 (1% w/v), sodium chloride (0.5% w/v), and HEPES (16.8 mM). Although the requirements of *F. nucleatum* do not allow us to do single-carbon–source experiments, this simple medium was used in a similar manner with or without added carbohydrate sources and is referred to throughout the text as "nutrient-limited" media.

### Identification of putative sialate lyase (NanA) homologs in *F. nucleatum*

A sequence similarity search was conducted using the protein blast (BLASTp ver. 2.2.31+) program from the BLAST suite [82]. Amino acid sequences of NanA from *E. coli* MG1655 (GenBank: AAC76257.1) and *F. nucleatum* subsp. *nucleatum* ATCC23726 (GenBank: EFG95907.1) were used as query to the nonredundant (NR) protein sequences database for *F. nucleatum* (taxid: 851). Any hits with an *E*-value less than $1 \times 10^{-9}$ were considered significant.

### Free sialic acid foraging by *F. nucleatum*

*E. coli* (Top 10) was cultured aerobically in LB medium with added sialic acid (Neu5Ac, 100 μM). Fusobacteria (S1 Table) were cultured anaerobically in supplemented Columbia

media with added sialic acid (Neu5Ac, 100 μM, A0812; Sigma-Aldrich, St. Louis, MO, USA), which was also used as the uninoculated medium control in this experiment. After 24–48 h of growth, media supernatant was separated from bacterial cells by centrifugation at $15,000 \times g$. In some experiments, collected media supernatant was filtered using a 10,000 molecular weight cutoff (MWCO) filter. Samples were then processed for 1,2-diamino-4,5-methylene-dioxybenzene (DMB) derivatization for measurement of free sialic acid (Neu5Ac) remaining in the medium, as described below.

## Measurement of total and free sialic acids by DMB-HPLC

Derivatization and quantitation of sialic acids by HPLC was carried out as previously described [31,83–86]. Briefly, total sialic acids were measured by first releasing any bound sialic acids using mild acetic acid hydrolysis (2 N acetic acid for 3 h at 80˚C), followed by derivatization of total sialic acids with DMB as described below. Alternatively, free sialic acid levels were measured by DMB derivatization without prior acid hydrolysis. The derivatization reaction requires that sialic acids be in the monosaccharide form. Thus, the concentration of bound sialic acids is measured indirectly (bound = total − free). Reaction conditions for DMB derivatization were 7 mM DMB, 22 mM sodium thiosulfite, 0.75 M 2-mercaptoethanol, and 1.4 M acetic acid for 2 h at 50˚C. Derivatized samples were injected into a Waters HPLC (Milford, MA, USA) equipped with a reverse-phase C18 column (Tosoh Bioscience, Tokyo, Japan) and eluted using isocratic conditions at 0.9 ml/min using 8% methanol, 7% acetonitrile in water. An online fluorescence detector (Waters) was set to excite at 373 nm and detect emission at 448 nm. Peak integrations were used to quantitate sialic acid content by referencing a standard curve of pure sialic acid (Neu5Ac; Sigma-Aldrich) derivatized in parallel.

## Complementation of *E. coli* MG1655 sialate lyase mutant *ΔnanA* and growth comparison

MG1655 *nanA* was amplified with the primers MG1655 nanA F Nco and MG1655 nanA R Pst (S1 Table). The PCR product was desalted, digested with the restriction enzymes NcoI and PstI, and ligated into the NcoI and PstI sites of pTrc99A to create pLR7. *F. nucleatum* ATCC23726 *nanA* was amplified with the primers Fuso nanA F Nco and Fuso nanA his R Bam. Following desalting and digestion with NcoI and BamHI, the amplicon was cloned into the same sites in pTrc99A, creating pLR10. The *E. coli* MG1655 *ΔnanA* mutant LSR4 was described previously [36]. LSR4 was transformed with pLR7, pLR10, or the vector control (pTrc99A) (S1 Table). The 3 strains were grown to an $OD_{600}$ of 1.0 in LB broth containing 100 μg/mL ampicillin. Then, each strain was diluted 1:100 into M63 media lacking glycerol and with added Neu5Ac (15 mM). The $OD_{600}$ was measured every 10 min at 37˚C in a 96-well plate (flat bottom; Greiner Bio-One, Monroe, NC, USA) in a TECAN M200 plate reader with 10 s of orbital shaking at the beginning of each time point.

## Sialate lyase assay by complementation of mutant *E. coli*

The *E. coli ΔnanA* mutant (LSR4) containing pLR7 (*E. coli* MG1655 *nanA*), pLR10 (*F. nucleatum nanA*), or empty vector control (pTrc99A) was grown by shaking at 37˚C in 35 mL Circle Grow broth with 100 μg/mL ampicillin to an $OD_{600}$ of 1.0. Cultures were then transferred to room temperature, induced with 0.2 mM IPTG, and grown by shaking overnight. By the next morning, cultures had grown to an $OD_{600}$ of approximately 4.0. Next, 25 ml of each culture was centrifuged at $12,000 \times g$ for 10 min, and the cells were resuspended in 800 μL PBS. Cell suspensions were sonicated on ice for 40 s in 1-s bursts at 24% amplitude in a Sonic Dismembrator (Thermo Fisher Scientific, Waltham, MA, USA). Insoluble debris was removed by

centrifugation at $15,000 \times g$ for 10 min, and 90 μL of each supernatant was mixed with 10 μL of 1 mM Neu5Ac for a final concentration of 100 μM. Mixtures were incubated at 37°C, and at each time point, 10-μL samples were removed, diluted 5× in water, and frozen at −20°C prior to analysis by DMB-HPLC.

## Isolation of an Sm^R mutant of *F. nucleatum* for selective enumeration

*F. nucleatum* ATCC23726 was cultured anaerobically overnight in supplemented Columbia media at 37°C. Cultured bacteria (20 mL) were pelleted by centrifugation at $16,000 \times g$ for 10 min at room temperature, resuspended in 1 mL PBS, and plated on supplemented Columbia blood plates with 1 mg/mL streptomycin sulfate to select for spontaneous Sm^R colonies. After 2 days, Sm^R colonies from Columbia plates (with strep) were cultured in supplemented Columbia broth. The Sm^R strain was used for all experiments with the exception of Fig 1A. Mutation of the *rpsL* gene (encoding the ribosomal protein S12) in the *F. nucleatum* ATCC23726-Sm^R strain was confirmed by colony PCR and DNA sequencing.

## Mutagenesis targeting sialic acid consumption via *siaT* in *F. nucleatum* ATCC23726

Restriction endonucleases, Phusion DNA polymerase, and T4 ligase were obtained from New England Biolabs (Ipswich, MA, USA). First, a suicide vector was constructed using the origin of replication from pUC19 and the chloramphenicol acetyltransferase gene (*catP*) from pJIR418 (S1 Table). Briefly, the origin of replication was amplified from pUC19 using primers pUC19 ori F Sac Kpn Nco and pUC19 ori R Bgl2. The *catP* gene from pJIR418 was amplified with primers catP F Bgl2 and catP R Bam Pst Nco. The 2 amplicons were desalted, digested with NcoI and BglII, and ligated together, resulting in pLR23. A 0.5-kb internal fragment of *F. nucleatum siaT* was amplified from ATCC23726 chromosomal DNA with the primers 23726 nan F Nco and 23726 nanR Sac. Following desalting, this PCR product and pLR23 were digested with NcoI and SacI and ligated together, creating pLR25. The Sm^R strain of *F. nucleatum* ATCC23726 was used as the background for the *siaT* mutation. A starter culture of this strain was diluted 1:1,000 into 10 mL supplemented Columbia broth and grown overnight in the anaerobic chamber at 37°C. Bacteria were then pelleted at $12,000 \times g$ for 10 min and washed twice in 1 ml of 1 mM MgCl$_2$ containing 10% glycerol. The bacteria were washed twice more in 1 mM MOPS with 20% glycerol and resuspended in 100 μl of the same buffer. Next, 20 μg of purified pLR25 was mixed with the above suspension and incubated on ice for 10 min. The suspension was then transferred to a cuvette with a 0.1-cm gap and electroporated in an Eppendorf Electroporator 2510 (Eppendorf, Enfield, CT, USA) set to 2.5 kV. Bacteria were recovered anaerobically for 5 h in supplemented Columbia broth with added MgCl$_2$ (1 mM) at 37°C, then plated on supplemented Columbia agar with 5 μg/mL thiamphenicol. Plasmid integration at the desired locus was confirmed by colony PCR using the primers 23726 nan test F1 and pLR24 5′ F. The mutant strain is referred to as *F. nucleatum* Ω*siaT*.

## RT-qPCR transcript analysis for genes flanking *siaT* in *F. nucleatum*

*F. nucleatum* WT and Ω*siaT* were grown anaerobically to exponential phase in supplemented Columbia broth at 37°C. RNA was isolated using PureLink RNA isolation kit (Ambion, Austin, TX, USA) according to the manufacturer's instructions and treated with 2.7 Kunitz units of DNase (Qiagen, Hilden, Germany) per microgram of RNA. cDNA was synthesized using a high-capacity cDNA Reverse Transcription kit (Thermo Fisher Scientific) following the manufacturer's protocol. Expression analysis at transcript level was done by RT-qPCR using Power SYBER Green master mix (Applied Biosystems, Foster City, CA, USA) with 1 ng of cDNA and

300 nM of each forward and reverse primer (S1 Table) in a final volume of 20 μL. Data were analyzed by the ΔΔCt method [87]. Expression of target genes was normalized to the relative expression of *F. nucleatum*-specific 16S rRNA [88]. Each experiment was performed in triplicate and repeated 3 times.

### Growth comparison of *F. nucleatum* WT and Ω*siaT* in Columbia broth

*F. nucleatum* ATCC23726 WT and Ω*siaT* were streaked out on supplemented Columbia blood plates with streptomycin (1 mg/mL) and grown in an anaerobic chamber at 37˚C for 24 h. Next day, colonies from the plates were gently scraped off using an inoculating loop and resuspended in supplemented Columbia broth. The $OD_{600}$ of the suspension was checked and adjusted to reach a final starting $OD_{600}$ of 0.10 in fresh media. Media alone or inoculated with *F. nucleatum* WT or Ω*siaT* (150 μL /well) was transferred to a clear flat-bottom 96-well plate (Cat. No. 655161; Greiner Bio-One) and sealed with a transparent seal (Cat No. 2978–2100; USA Scientific, Ocala, FL, USA) to maintain anaerobic conditions. Growth was monitored by recording the $OD_{600}$ every 10 min, with 10 s linear shaking (amplitude 3 mm) followed by a 1-min pause at each time point, at 37˚C in a TECAN M200 plate reader (Morrisville, NC, USA) for approximately 30 h.

### Growth comparison of *F. nucleatum* WT and Ω*siaT* in nutrient-limited media

Experiments were conducted in a nutrient-limited media containing 1% (w/v) proteose peptone #3, 0.5% (w/v) sodium chloride, and 16.8 mM HEPES under anaerobic conditions. *F. nucleatum* ATCC23726 WT and Ω*siaT* were first cultured overnight anaerobically in supplemented Columbia media. After approximately 16 h, in a clear flat-bottom 96-well plate (Cat. No. 655161; Greiner Bio-One), each strain of *F. nucleatum* was inoculated in nutrient-limited media (200 μL/well) containing different added carbohydrate sources as follows: (i) 30 mM glucose, (ii) no added carbon source (low-carbohydrate), (iii) 30 mM Neu5Ac, (iv) 30 mM 3SL (αNeu5Ac2-3βDGal1-4DGlc), and (v) 30 mM 3SL with 50 mU of *A. ureafaciens* sialidase. Both strains were inoculated at starting $OD_{600}$ of 0.1, and the plate was sealed with a transparent seal (Cat No. 2978–2100; USA Scientific) to maintain anaerobic conditions. Growth was monitored by recording OD at 600 nm every 10 min, with 10 s linear shaking (amplitude 3 mm) followed by a 1-min pause at each time point at 37˚C in a TECAN M200 plate reader for approximately 50 h.

### Fluorometric assay for sialidase activity

All sialidase assays were performed in 96-well U-bottom black polypropylene plates (Eppendorf). Plates were sealed with an optical clear film (Cat No. 2978–2100; USA Scientific) and substrate hydrolysis was monitored by measuring fluorescence of 4-methylumbelliferone (4MU) (excitation 365 nm, emission 440 nm) in a TECAN M200 plate reader every 2 min for 1–2 h at 37˚C. To test for potential sialidase activity produced by *G. vaginalis* and *F. nucleatum*, overnight cultures were adjusted to an $OD_{600}$ of 0.5 and 25 μL was mixed with an equal volume of PBS in the plate, followed by the addition of 100 μL Dulbecco's PBS containing 100 μM 4MU-Neu5Ac (4MU-sialic acid; Goldbio, St. Louis, MO, USA). To monitor sialidase activity in mouse vaginal washes, 25 μL of vaginal wash was mixed with 50 μL PBS containing 100 μM 4MU-sialic acid. A standard curve of 4MU in PBS (0–67 μM) was included in parallel to calculate the rate of 4MU-sialic acid hydrolysis by comparison to standards. Rate of 4MU-sialic acid hydrolysis was used to compare sialidase activity in the mouse vaginal washes between different time points or over a period of time. For in vitro experiments examining the effect of *F*.

*nucleatum* on cultured mouse/human vaginal specimens or *G. vaginalis*, 50 μL of the final culture was mixed with 100 μL sodium acetate buffer (pH 5.5) containing 500 μM 4MU-sialic acid. Similarly, for sialidase activity assays on (uncultured) human vaginal specimens, 50 μL of resuspended swab eluates was transferred to the plate and mixed with 100 μL sodium acetate buffer (pH 5.5) containing 500 μM 4MU-sialic acid. 4MU in sodium acetate buffer (0–300 μM) was used as a standard in sialidase assays with vaginal swab eluates.

## Foraging of sialic acids from bound sialoglycans in nutrient-limited media by *F. nucleatum* in the presence or absence of exogenous sialidase

*F. nucleatum* WT and Ω*siaT* were cultured in nutrient-limited media containing 3SL with or without *A. ureafaciens* sialidase, as described above. At the end of the growth experiment, 96-well plates were centrifuged at $3,000 \times g$ for 20 min at 4˚C to separate the media from bacterial cells. Thereafter, spent supernatant from each growth condition (approximately 150 μL/well) was transferred to 0.2-μm 96-well filter plates (Pall Corporation, Port Washington, New York, USA) stacked onto a v-bottom 96-well collection plate (Eppendorf) and centrifuged at $3,000 \times g$ for 15 min at 4˚C. Total and free sialic acids remaining in the filtered supernatants were measured by DMB labeling as described earlier.

## Mouse vendor information, housing conditions, and general handling procedures

For experiments with *F. nucleatum*, female C57BL/6 mice (6–7 weeks old) were obtained from Jackson Labs (Bar Harbor facility, MA, USA) and Envigo (202-A Indianapolis facility, IN, USA). Mice were received between January 2016 and January 2017. For experiments testing vaginal sialidase activity among different mouse vendors, female C57BL/6 mice were also obtained from Charles River (NCI Grantee, Frederick facility, MD, USA) and Taconic. Once received at Washington University, mice were housed in a barrier facility kept at $70 \pm 2$˚F, with a 12:12 light/dark cycle, and using corncob bedding, PicoLab Rodent Diet 20, and nestlets for enrichment changed at least once per week. During initial experiments, mice were housed in standard (nonventilated) microisolator cages. Later, our animal facility changed to a new housing system in which larger cages are docked in HEPA-filtered ventilation racks that provide airflow control. Mice were allowed to rest for approximately 1 week prior to the commencement of procedures. Mice were estrogenized as indicated in figure legends by intraperitoneal injection under isoflurane anesthesia with 0.5 mg β-estradiol 17-valerate (E1631; Sigma-Aldrich) in 100 μL filter-sterilized sesame oil. Vaginal washes were collected from isoflurane-anesthetized animals by gently inserting a pipette tip with 50 μL of sterile Dulbecco's PBS approximately 2–5 mm into the vagina and pipetting up and down.

## Mouse model of vaginal colonization with *F. nucleatum*

Our spontaneous Sm$^R$ isolate derived from ATCC23726, which we refer to simply as WT *F. nucleatum*, was vital to these experiments because it allows us to distinguish inoculated *F. nucleatum* from endogenous members of the microbiota for enumeration of CFUs. The Ω*siaT* strain described earlier was derived from this Sm$^R$ WT strain. Mice were injected twice with β-estradiol (see above) 3 days prior to and on the day of inoculation, with all animals receiving the first dose approximately 1 week postarrival. To compare the vaginal colonization of *F. nucleatum* WT versus Ω*siaT*, a total of 80 mice (40 mice from each vendor) were used in 2 independent experiments, conducted approximately 1 year apart, with each individual experiment including 20 mice from Envigo Labs and 20 from Jackson Labs. Ten mice per vendor

were vaginally colonized with *F. nucleatum* WT and 10 with the *F. nucleatum* Ω*siaT*. To prepare the *F. nucleatum* inoculum, the overnight culture was adjusted to an $OD_{600}$ of 4.0 in supplemented Columbia broth and aliquoted in 1.5 mL Eppendorf tubes (25 μL/ tube) for each individual mouse. Mice were vaginally inoculated with approximately $10^8$ CFUs of WT or Ω*siaT* mutant in 20 μL supplemented Columbia broth using a separate aliquot of inoculum for each mouse. In a separate experiment to compare persistence of sialidase activity, Envigo mice were either mock treated by inoculating with supplemented Columbia broth (vehicle) or WT *F. nucleatum* in supplemented Columbia broth ($n$ = 10 per group). Vaginal washes were collected every 2–3 days, and mice were considered sialidase-negative as of the first of 3 sequential time points in which sialidase activity levels in vaginal washes were below 0.01 μM/min using the 4MU-Sia assay.

Vaginal washes were collected one day before estradiol was first administered, after estrogenization (on the day of inoculation), and then 3 times per week (on alternate days) until 38 dpi or 44 dpi (for mock versus *F. nucleatum*). Mice were considered "cleared" of *F. nucleatum* when 2 consecutive washes were negative for *F. nucleatum* CFUs. Once cleared, mice did not reacquire *F. nucleatum*, even if cohoused with colonized mice. Washes were collected from all mice by flushing vaginas with 50 μL sterile PBS and rinsing into an additional 20 μL PBS in a sterile 1.5-mL Eppendorf tube. All washes were collected (from 40 mice) within 1 h and cycled into the anaerobic chamber for CFU enumeration. *F. nucleatum* titers were determined from washes by preparing 10-fold serial dilutions in PBS and spotting 5 μL of each dilution in quadruplicate onto supplemented Columbia selection plates with 1 mg/mL streptomycin. After anaerobic incubation at 37°C, colonies were then enumerated and reported as recovered CFU per mL of vaginal fluid. Sialidase activity was measured in freshly collected mouse vaginal washes as described earlier. At appropriate experimental endpoints, mice were sacrificed by cervical dislocation under isoflurane anesthesia or by using carbon dioxide at 10%–30% chamber volume displacement per min using a SmartBox.

### Ex vivo mouse vaginal community cultures

Mouse vaginal wash material remaining after sialidase assays and *F. nucleatum* CFU analysis was pooled by cage (5 mice per cage) for each vendor or treatment condition. A small portion (5 μL) of each pooled vaginal wash sample was used as inoculum to culture the vaginal bacterial communities in supplemented Columbia broth under anaerobic conditions. We refer to these vaginal communities as "microbiota pools." Each microbiota pool represents either uncultured pooled vaginal wash or cultured bacterial community from pooled vaginal wash of mice cohoused in the same cage. The cultured communities (first passage) were frozen with 20% glycerol after overnight growth. Remaining pooled vaginal washes were stored at −20°C for later microbiome studies (see below). See also S1 Schematic.

### *F. nucleatum*'s influence on sialidase activity in mouse vaginal communities

Experiments were done with microbial communities cultured using pooled vaginal wash specimens (referred to as "microbiota pools") collected from both Envigo and Jackson mice after estrogenization but before inoculation with *F. nucleatum*. These experiments were done in Columbia broth, a culture medium that contains sialoglycans from animal ingredients, but these remain inaccessible to *F. nucleatum* because of lack of sialidase activity. Cultured mouse vaginal communities (first passage) frozen in Columbia media were streaked out on Columbia blood plates and grown in an anaerobic chamber at 37°C (second passage). After 2 days, Columbia blood plates were gently scraped using an inoculating loop, and bacteria were resuspended in supplemented Columbia broth. The $OD_{600}$ of the suspension was checked and

adjusted to 0.5 for the vaginal communities and 1.0 for both *F. nucleatum* strains. Experiments were done in v-bottom, 96-well, sterile, deep-well (2-mL) plates (Eppendorf). *F. nucleatum* WT or Ω*siaT* was added in equal amounts to 700 μL (total volume of media per well) of Columbia broth at a starting $OD_{600}$ of either 0.10 or 0.025, followed by addition of vaginal communities at starting $OD_{600}$ 0.05. Appropriate controls such as uninoculated Columbia broth, Columbia broth inoculated with *F. nucleatum* alone at a starting $OD_{600}$ of 0.10, and Columbia broth inoculated with each individual vaginal microbial community alone were included in each experiment. After the addition of aluminum foil seals (Product ID: BK538619; Beckman Coulter Life Sciences, Indianapolis, IN, USA), plates were incubated O/N (approximately 16 h) at 37˚C in the anaerobic chamber. After O/N growth (third passage of mouse communities), sialidase activity was checked in each well using the fluorometric siali-dase assay as described earlier in the Methods. Also, growth of *F. nucleatum* WT and Ω*siaT* with the microbiota pools was determined at this point by preparing 10-fold serial dilutions of the O/N cultures in PBS and spotting 5 μL of each dilution onto Columbia selection plates con-taining 1 mg/mL streptomycin and 5 mg/L of crystal violet. Microbiota pools cultured without *F. nucleatum* did not show any growth on these plates (crystal violet selects against gram-posi-tive bacteria), thus allowing selective enumeration of *F. nucleatum* in the cocultures. See also S1 Schematic.

## DNA extraction and community profiling of mouse vaginal washes

For microbiome analysis by 16S sequencing, pooled vaginal washes (uncultured microbiotas) from 8 different cages (4 for each vendor, 1 specimen per cage) were centrifuged at $16,000 \times g$ for 5 min, the supernatant was removed, and the pellet was used for DNA extraction. For cul-tured vaginal communities (4 for each vendor, 1 specimen per cage), communities frozen in supplemented Columbia broth were streaked out on supplemented Columbia blood plates and grown overnight at 37˚C in an anaerobic chamber (second passage). Colonies from plates were gently scraped off using a cell scraper and resuspended in supplemented Columbia broth. For DNA extraction, bacterial cells from these communities were separated from the superna-tant by centrifugation at $16,000 \times g$ for 5 min. DNA was extracted using the Wizard Genomic DNA Purification kit from Promega (Cat. No. A1120; Madison, WI, USA) following supplier instructions. See also S1 Schematic.

## Mouse vaginal community analysis by V1–V2 sequencing of the 16S rRNA gene

For the mouse microbiome experiments, we decided to use the 16S V1–V2 regions after doing a preliminary experiment using the 16S V4 region, which was unable to identify and distin-guish members of the *Lactobacillales* down to the species level (for example, *E. faecalis* and *E. casseliflavus/gallinarum*). We amplified the V1–V2 region of the 16S ribosomal subunit with the universal 27F and 338R primers [4]. Both the primers contained a common adaptor sequence and the forward primer also contained a barcode sequence for multiplexing. The primers were as follows: 338R-5′-AGACGTGTGCTCTTCCGATCTCAT**GCTGCCTCCCG TAGGAGT**-3′ and 27F-5′-ACGACGCTCTTCCGATCTNNNNNNNNCT**AGAGTTTGAT CCTGGCTCAG**-3′, with the bolded sequences denoting the universal primers and 8-bp bar-code being denoted by 8 Ns. After the V1–V2 region was amplified using the 27F and 338R primer pair, the PCR product was treated with Exo-SAP-IT to remove primers. A second PCR was performed to add unique indexes for further multiplexing. The subsequent amplicons were quantified and pooled. The pool was run on a 0.8% agarose gel, excised, and extracted. Agencourt AMPure XP beads (Beckman Coulter Life Sciences) were used for further size

selection and purification. The $2 \times 150$ paired-end sequencing was completed using the Illumina MiniSeq platform (San Diego, CA, USA) through the Center for Genome Science at Washington University School of Medicine. Reads were trimmed for adaptor sequences using ea-utils fastq-mcf version 1.04.676 and de-multiplexed based on the unique barcode and index identifiers. USEARCH [89] version 10.0.240 was used for the following USEARCH commands: (i) reads were merged using the fastq_mergepairs command and (ii) quality filtered with a maximum expected error of 1 and a minimum sequence length of 100 using the fastq_filter command, (iii) merged reads were then dereplicated using the fastx_uniques command, and (iv) OTUs were clustered using the cluster_otus command (97% OTU clustering using UPARSE-OTU algorithm) or unoise3 command (UNOISE algorithm also performs denoising of amplicon reads). As part of the cluster_otus and unoise3 commands, chimeric sequences were filtered out and discarded, and a minimum abundance of 2 reads per OTU was applied. OTUs were assigned taxonomic predictions using the RDP 16S database (version 16) with a confidence threshold of 0.7. For normalization, samples were rarified to 1,000 reads per sample. To quantify relative abundance of each OTU in a given sample, OTU reads were log transformed using the following equation: log([reads + 1]/total reads). All OTUs at >1% relative abundance in at least one sample were included in analysis. Data were clustered using hierarchical Euclidean clustering in the R project for statistical computing. As in other microbiome studies, sampling methods and primer choice could bias the types of microbes recovered and detected. These limitations may apply to the vaginal microbiome in mice as well as in women.

### Mouse model of vaginal inoculation with *E. coli*

Female C57BL/6 mice, 5–7 weeks old, were obtained from Envigo (202-A Indianapolis facility). At 48 and 24 h prior to infection, mice underwent intraperitoneal injections of 0.5 mg β-estradiol 17-valerate (Sigma-Aldrich) in 100 μl filter-sterilized sesame oil (Thermo Fisher Scientific) to synchronize the mice in estrus. Mice were infected with model uropathogenic *E. coli* strains (UTI89 and CFT073 [90,91]) or mutant derivatives thereof. Strains were cultured statically in lysogeny broth (LB) at 37˚C for 2 consecutive overnight passages. At the time of infection, $10^4$ CFUs of each *E. coli* strain were inoculated into the vaginas in 20-μl volumes of PBS. To monitor infection and sialidase status, vaginal washes were collected as described above, serially diluted, and plated on MacConkey agar (titers of these uropathogenic strains went up to approximately $10^8$ CFUs by 24 hpi) [59]. Sialidase activity was measured in freshly collected mouse vaginal washes as described above.

### Study design for human subject research

To test our hypothesis that sialidase activity would significantly increase in living communities cultured with *F. nucleatum* compared with identical communities exposed only to vehicle, our experimental design required that some detectable level of sialidase be present within the microbiome of tested communities. Sialidase activity was measured in 51 available (standard aerobic) swab specimens, of which approximately 60% were sialidase-positive. For the power calculation, our primary outcome was sialidase activity after overnight culture of cryogenically preserved living communities. The exact Agresti–Coull method was used to estimate the sample size needed for the ex vivo studies. To achieve 80%–90% power (alpha = 0.05) to detect sialidase altered in 80% of samples (versus the theoretical random distribution of sialidase levels, which would give 50% of samples with higher sialidase activity in the treatment group), we calculated that we would need 22–28 independent communities. Our experimental design in 96-well plates using different inocula of *F. nucleatum*

allowed us to evaluate a total of 21 communities in 2 independent experiments (11 communities in the first experiment and 10 communities + 1 duplicate community in the second). There was good agreement between the data for the community that was evaluated in duplicate. Combined data from both experiments are shown (each data point represents a community from 1 individual). Further measurements of *G. vaginalis* levels (by qPCR and proportions within the postculture communities) were performed as described in the Methods below.

## Collection and processing of human vaginal swabs

Midvaginal swabs were collected during speculum exam by a clinician, immediately submerged into prereduced Cary Blair media using Starswab Anaerobic Transport System S120D (Starplex Scientific, Ontario, Canada), and transported to the laboratory for same-day processing. Once transported anaerobically to the lab, vaginal swabs were cycled into the anaerobic chamber immediately. Swabs were eluted in 2× NYCIII media without glucose, and "fresh-frozen" (i.e., "0 passage," without growth or amplification of any kind) in the presence of sterile glycerol (20% final). In parallel with clinical specimens returning to the lab in anaerobic transport devices, control swabs exposed to the air in clinic exam rooms were also transported anaerobically to the lab and processed in parallel with clinical specimens. Standard (aerobic) double-headed rayon swabs S09D (Starplex Scientific) collected at the same visit were eluted in 1 mL of pH 5.5 sodium acetate buffer, and sialidase activity was measured as described earlier in the Methods. For coculture experiments with *F. nucleatum*, we selected anaerobic fresh-frozen vaginal microbial communities from women whose aerobic vaginal swab eluates had detectable sialidase activity (S7 Fig). Fresh-frozen communities from 21 sialidase-positive women were used. See also S2 Schematic.

## Ex vivo studies of human vaginal communities

*F. nucleatum* ATCC23726 WT was streaked out on supplemented Columbia blood plates and grown in an anaerobic chamber at 37°C for 24 h. The next day, *F. nucleatum* colonies from plates were gently scraped off using an inoculating loop and resuspended in supplemented Columbia broth. The $OD_{600}$ of the suspension of *F. nucleatum* was adjusted to 1.0. Then, *F. nucleatum* inocula were prepared by 2-fold serial dilution in supplemented Columbia media. Fresh-frozen (passage 0) human vaginal communities ($N$ = 21) were thawed on ice and diluted 4-fold in 1× NYCIII with no added glucose (low-carbohydrate medium). Experiments were done in v-bottom, 96-well, sterile, deep-well (2-mL) plates (Cat. No. 951033502; Eppendorf). *F. nucleatum* was inoculated at a starting $OD_{600}$ of 0.10, followed by addition of 116 μL of the diluted fresh-frozen community in a total of 700 μL media (supplemented Columbia broth) per well. Mock vaginal communities were included as controls in each experiment; these mock communities were originally prepared from blank swabs that were eluted and stored using the same media and reagents, in parallel with swabs returning from the clinic. The plates were sealed with a sealing aluminum foil (Product ID: BK538619; Beckman Coulter) and incubated overnight (17 h) at 37°C in the anaerobic chamber. After overnight growth (first passage of human communities), 50 μL of culture from each condition was transferred to round-bottom black plates, and sialidase activity was measured in each well using the fluorometric sialidase assay as described earlier in the Methods. The 96-well plates with the remaining cultures were centrifuged at $3,200 \times g$ for 20 min at 4°C, and supernatants were removed carefully without touching the bacterial cell pellets at the center of the well and stored at −80°C for DNA extraction and PCR as described below. See also S2 Schematic.

## DNA extraction from ex vivo-cultivated human vaginal communities

DNA was extracted from bacterial cell pellets, saved from the *F. nucleatum* human vaginal communities coculture experiments described above, by bead beating followed by phenol chloroform extraction. Briefly, a 96-well (deep-well, v-bottom) plate with bacterial pellets was thawed, and cells were resuspended in 250 µL of buffer containing 200 mM Tris-HCl (pH 8.0), 200 mM NaCl, and 20 mM EDTA. This suspension was transferred to sterile 2-mL tubes (Axygen, Corning, NY, USA) containing approximately 250 µL silica beads and 105 µL 20% SDS, followed by addition of 250 µL phenol/chloroform/isoamyl alcohol saturated with 10 mM Tris (pH 8.0). After bead beating for 3 min, tubes were centrifuged at 3,200 × *g* for 4 min, and the crude DNA was transferred to clean up columns (Qiagen). Concentration of the purified DNA was determined using Quant-iT BR dsDNA Assay Kit (Thermo Fisher Scientific). Before PCR, DNA was normalized to 2.5 ng/µL for each condition.

## 16S profiling of ex vivo-cultivated human vaginal communities

16S rRNA PCR was performed in triplicate for each template using Platinum Taq DNA Polymerase, High Fidelity (Thermo Fisher Scientific) using barcoded V4-specific primers as previously described [92]. No-template controls were additionally run with every barcoded primer to ensure that reactions were free from contamination. Replicate PCR reactions were combined and then quantitated using Qubit dsDNA Broad Range Assay reagents (Invitrogen, Waltham, MA, USA). Quantitated DNA V4 16S reactions were then pooled into a single library in an equimolar ratio using the epMotion 5075 TMX (Eppendorf, Hamburg Germany). V4 16S rRNA amplicon libraries were then sequenced on an Illumina MiSeq instrument using 2 × 250 base pair chemistry. Amplicon sequence variants were generated from the de-multiplexed data using the large-data–adapted pipeline for DADA2 version 1.10.1 in R (https://benjjneb.github.io/dada2/bigdata.html) [93]. The forward and reverse reads were trimmed to 230 base pairs and 160 base pairs, respectively, based on the average quality profile of forward and reverse reads. Sequences with more than 2 likely erroneous bases or any ambiguous bases were then filtered out. Sequencing errors were learned and amplicons were then de-replicated and partitioned into amplicon sequence variants as described [93]. Forward and reverse reads were subsequently merged, and chimeras were removed. Amplicon sequence variant sequences were assigned taxonomy from kingdom to species using the DADA2 inbuilt function for the naïve Bayesian Classifier [94] with a custom database [95] and a minimum bootstrap support of 80%. For data shown in Fig 7C, we excluded the reads for *F. nucleatum* ATCC23726 (added exogenously) when determining the total reads to calculate relative levels of *G. vaginalis*. Some samples were excluded from the final analysis that had insufficient numbers of reads (0) mapping to *G. vaginalis* in either of the conditions tested.

## Cloning the *tuf* gene from *G. vaginalis*

The primer pair Gvag_tuf_AS3/Gvag_tuf_S4 was used to amplify a 149-bp fragment from the *tuf* gene (GI:311114364) of *G. vaginalis* strain JCP8481B. The PCR product was purified using the Nucleospin Gel and PCR Clean-Up Kit (Clontech, Takara Bio, Mountain View, CA, USA) and cloned into the Zero Blunt TOPO vector (Invitrogen) according to the manufacturer's instructions.

## Quantification of *G. vaginalis* by *tuf* qPCR

DNA was extracted from human vaginal communities cultured with/without *F. nucleatum* as described above. qPCR was performed using PowerUp SYBR Green Master Mix (Applied

Biosystems) with 5 ng of DNA and 250 nM of *tuf* forward and reverse (Gvag_tuf_AS3/Gvag_-tuf_S4) primers in a final volume of 20 μL. Plasmid containing a 149-bp fragment of *tuf* gene from *G. vaginalis* was employed to generate a standard curve for quantitation. Three replicates were included for each experimental condition, including the no-template control, DNA derived from media-only controls, and blank swab controls. The average CT values from the triplicate reactions were used to calculate the *tuf* copy numbers using the linear equation derived from the standard curve. No-template controls, media-only controls, and blank swab controls fell outside the standard curve (less than 800 copies). *G. vaginalis* levels are expressed as *tuf* gene copies per community.

## Coculture experiments with *Fusobacterium* and *G. vaginalis*

Coculture experiments were done in v-bottom, 96-well, sterile, deep-well (2 mL) plates (Eppendorf). *F. nucleatum* ATCC23726 (WT or Ω*siaT*) and *G. vaginalis* JCP8151B (Sm$^R$) were streaked out on supplemented Columbia blood plates and NYCIII plates with streptomycin (1 mg/mL), respectively, and grown in anaerobic chamber at 37˚C for 24 h. The next day, colonies from the plates were gently scraped off using an inoculating loop and resuspended in supplemented Columbia broth. For experiments measuring sialidase activity in the coculture, the OD$_{600}$ of the suspension was checked and adjusted to 0.05 ($1.20 \times 10^8$ CFU/mL) for *G. vaginalis* and 0.1 for *F. nucleatum* (WT, $1.80 \times 10^8$ CFU/mL; Ω*siaT*, $2.20 \times 10^8$ CFU/mL). A series of inocula were prepared for each strain by 2-fold serial dilution. For coculture, prepared inocula for both *F. nucleatum* and *G. vaginalis* were diluted 10-fold in a total of 700 μL of supplemented Columbia broth per well. The plate was sealed with a sealing aluminum foil (Beckman Coulter) and incubated overnight (17 h) at 37˚C in the anaerobic chamber. After overnight growth, sialidase activity was measured in each well using the fluorometric sialidase assay as described earlier. *G. vaginalis* titers were determined after overnight growth for a subset of the wells (Fig 8A, *G. vaginalis* inoculum $2.28 \times 10^5$ CFU/mL), with and without WT *F. nucleatum* (at the indicated inoculum OD), by preparing 10-fold serial dilutions in PBS (in the anaerobic chamber) and spotting 5 μL of each dilution in quadruplicate onto NYCIII plates containing 1 mg/ml streptomycin, 10 μg/ml colistin, and 10 μg/ml nalidixic acid (colistin and nalidixic acid select against gram-negative bacteria and allow the specific enumeration of *G. vaginalis* in this assay). In separate experiments, *G. vaginalis* was cocultured with or without *F. nucleatum* (inoculum OD$_{600}$ 0.10) at varying inocula, with the highest *G. vaginalis* inoculum tested having OD$_{600}$ 1.3 ($1 \times 10^9$ CFU/mL), and titers were determined after overnight growth as described above.

## Experiments with *F. nucleatum* cell-free supernatants

Preparation of cell-free supernatants: *F. nucleatum* WT inoculum was prepared as described earlier. A 10 mL culture was started in supplemented Columbia broth at OD$_{600}$ 0.01 ($2.8 \times 10^7$ CFU/mL of *F. nucleatum*). After overnight growth (approximately 18 hr), bacterial cells were pelleted by centrifugation at $12,000 \times g$ for 10 min, and the supernatant was transferred to a new tube. A second centrifugation step was included ($12,000 \times g$ for 10 min) to remove any remaining bacterial cells, and the supernatant was subsequently filtered through a 0.22-μm membrane to obtain cell-free spent media. These filtered supernatants were spread plated on supplemented Columbia agar and incubated at 37˚C for 2 days (no live bacteria were recovered, LOD = 10 CFU/ml). The cell-free supernatant was cycled in the anaerobic chamber at least 4 h before inoculation with *G. vaginalis*.

Growth of *G. vaginalis* in cell-free supernatants from *F. nucleatum*: Experiments were done in v-bottom, 96-well, sterile, deep-well (2-mL) plates (Cat. No. 951033502; Eppendorf).

Inocula of *G. vaginalis* ($3.0 \times 10^5$ CFU/mL) and *F. nucleatum* ($OD_{600}$ 0.1, $2.5 \times 10^8$ CFU/mL) were prepared from colonies on solid-media plates (after 24 h growth) as described above. The inoculum was diluted 10-fold in supplemented Columbia broth with or without cell-free supernatants in a final volume of 700 μL medium per well. The plate was sealed and incubated overnight (approximately 18 h) at 37°C in the anaerobic chamber. After overnight growth, CFUs were enumerated by 10-fold serial dilutions in PBS as described above.

## Statistical analysis

R project for statistical computing was used for hierarchical clustering of microbiome data. GraphPad Prism 7.0 software was used for all statistical analyses presented. The statistical tests used to analyze each set of data and the number of replicates or independent experiments are indicated in the figure legends. For animal experiments, the figures with titers or sialidase activity show individual data, with each data point representing a value from a different animal with a line at the median. For experiments with human vaginal communities, each data point represents a vaginal community from an individual woman. For nonparametric analyses, differences between the experimental groups were analyzed with a two-tailed unpaired Mann–Whitney U test, or, when making multiple comparisons, we used Dunn's test. In the experiments in which data sets are paired, differences were analyzed using Wilcoxon matched-pairs sign rank test. Survival analysis was done using the Gehan–Breslow–Wilcoxon test.

## Supporting information

**S1 Fig. A subgroup of *F. nucleatum* strains encode putative sialate lyase (NanA) homologs.** (A) Heat map showing percent identity of NanA homologs in sequenced F. nucleatum strains to the amino acid sequence of sialic acid lyase of E. coli MG1655. Homologs showed high similarity to amino acid sequence of lyase from F. nucleatum ATCC23726. ATCC, American Type Culture Collection; NanA, N-acetylneuraminate lyase.
(TIFF)

**S2 Fig. Growth of *F. nucleatum* in supplemented Columbia media.** (A) F. nucleatum WT and ΩsiaT show similar growth when cultured anaerobically under these culture conditions. Data shown are representative of 3 independent experiments. siaT, predicted sialic acid transporter; WT, wild type.
(TIFF)

**S3 Fig. C57BL/6 mice from Envigo have endogenous vaginal sialidase activity.** (A) Sialidase activity in vaginal washes of mice (not estrogenized) from different vendors measured using the fluorogenic 4MU-Neu5Ac substrate. $N$ = 10 mice/vendor. (B) Vaginal sialidase activity is elevated at 72 h postestrogenization in mice from Envigo. No sialidase activity was detected in vaginal washes of mice from Jackson. $N$ = 40 mice/vendor. Wilcoxon paired-sign rank test was used for pairwise comparison. $^{**}P < 0.01$. (C) Sialidase activity in microbiota pools from Envigo and Jackson mice, collected after estrogenization. Each "microbiota pool" consists of a cultured vaginal community from pooled vaginal washes of 4–5 cohoused mice. For A, B, and C, data shown are combined from 2 independent biological replicates. (D) Heat map shows relative abundance of bacterial taxa identified by 16S rRNA V1–V2 sequencing in vaginal specimens of Envigo and Jackson mice collected before and after estrogenization. Microbiome analysis was done on uncultured and cultured vaginal washes pooled from 5 mice housed in the same cage. Each column represents 1 pool = 1 cage = 5 mice. Total = 4 microbiota pools per vendor per condition. OTUs were clustered using UPARSE-OTU algorithm, and taxonomic predictions were assigned using the RDP 16S rRNA database. (E) For identification of

sialidase-positive bacteria in Envigo mice, frozen microbiota pools were streaked on supplemented Columbia blood plates anaerobically incubating for 24 h at 37˚C. Single colonies were gently picked from these plates and cultured overnight. (F) Sialidase-positive cultures were identified using 4MU-Sia assay. Strains were identified by amplification and sequencing of 16S gene. Percent identity to the recovered blast hits is given in parentheses. The underlying numerical data for this figure can be found in S1 Data and S2 Data. Neu5Ac, *N*-acetylneuraminic acid; OTU, Operational Taxonomic Unit; RDP, Ribosomal Database Project; 4MU, 4-methylumbelliferone.
(TIFF)

**S4 Fig. *Fusobacterium* vaginal colonization and sialic acid foraging in C57BL/6 mice from Envigo inoculated with *F. nucleatum* WT and Ω*siaT*.** (A) *F. nucleatum* titers in vaginal wash collected at the indicated time points postinoculation. Data are combined from 2 independent experiments. Each experiment had 10 mice per group. $^*P < 0.05$, $^{**}P < 0.01$, Mann–Whitney. (B) Free and total sialic acid (Neu5Ac) concentrations in vaginal wash at 8 dpi. Total $N = 18$. The underlying numerical data for this figure can be found in S1 Data. dpi, days postinoculation; *N*-acetylneuraminic acid; *siaT*, predicted sialic acid transporter; WT, wild type.
(TIFF)

**S5 Fig. *F. nucleatum* WT and Ω*siaT* vaginal colonization in C57BL/6 mice from Jackson.** (A) *F. nucleatum* titers in vaginal wash collected at the indicated time points postinoculation. (B) Comparison of vaginal colonization with the WT versus Ω*siaT*. Number of mice colonized in percent (*y* axis) was monitored on day 1 and every 2 days thereafter for 38 days (*x* axis). For Kaplan–Meier analysis, mice were considered cleared when no CFUs were detected in undiluted wash at 2 consecutive time points. Data are combined from 2 independent experiments. Each experiment had 10 mice per group. Statistical significance assessed by the Gehan–Breslow–Wilcoxon test revealed no significant difference in colonization by WT versus Ω*siaT* in mice from Jackson. The underlying numerical data for this figure can be found in S1 Data. CFU, colony-forming unit; *siaT*, predicted sialic acid transporter; WT, wild type.
(TIFF)

**S6 Fig. Vaginal sialidase activity in *E. coli*- and *F. nucleatum*-inoculated mice.** (A–B) Sialidase activity in vaginal washes from individual animals purchased from Envigo, estrogenized, and inoculated with *E. coli* (A) or *F. nucleatum* (B) from 1 to 8 dpi. The underlying numerical data for this figure can be found in S1 Data. dpi, days postinoculation.
(TIFF)

**S7 Fig. Analysis of sialidase activity in vaginal swab eluates for selection of human vaginal communities.** (A) Graph shows relative sialidase activity in vaginal swabs that were transported aerobically from clinic to the lab. Sialidase activity was measured in swab eluates ($N = 58$) using a 4MU-Sia assay. Samples with a relative sialidase activity >1.0 were considered to be sialidase-positive. Red = sialidase-positive specimens, green = blank swab controls, blue = buffer control. The underlying numerical data for this figure can be found in S1 Data. 4MU, 4-methylumbelliferone.
(TIFF)

**S8 Fig. Sialidase activity amplification in *G. vaginalis* cocultured with *F. nucleatum*.** (A) Heat map shows relative sialidase activity in *G. vaginalis* (inoculum $OD_{600}$ 0.0–0.05) cultured overnight with *F. nucleatum* Ω*siaT* (inoculum $OD_{600}$ 0.0–0.1) in supplemented Columbia media. Data shown are representative of 2 independent experiments. OD, optical density;

*siaT*, predicted sialic acid transporter.
(TIFF)

**S1 Schematic. Mouse vaginal microbial communities—collection and amplification.** Step 1: collection of mouse vaginal washes. Publication of this animal image was approved by IACUC. Washes were pooled from mice cohoused in the same cage. A portion of vaginal wash pools was cultured in Columbia broth (referred to as "microbiota pools") to amplify the vaginal bacteria and frozen for subsequent use. Remaining pooled material was stored at −20°C. Step 2: on the day of the experiment, frozen microbiota pools were used to recover mouse vaginal bacteria by streaking out on supplemented Columbia blood plates in anaerobic chamber and incubating for 24 h at 37°C. Colonies from these plates were resuspended in liquid media either (a) for DNA extraction or (b) for coculture experiments with *F. nucleatum*. IACUC, Institutional Animal Care and Use Committee.
(TIFF)

**S2 Schematic. Human vaginal microbial communities—collection and amplification.** Step 1: anaerobic and aerobic vaginal swabs were collected on the same day from each participant. Aerobic swabs were eluted in sodium acetate buffer (pH 5.5), and sialidase activity was checked in the swab eluates using fluorogenic 4MU-Neu5Ac substrate. Anaerobic swabs were eluted in 2× NYCIII media (in an anaerobic chamber) and the communities were "fresh frozen," without any amplification/overnight culture, by mixing with cryoprotectant and storing at −80°C. Step 2: on the day of the experiment, fresh-frozen anaerobic vaginal communities, from women who had detectable sialidase activity in their aerobic swab eluates, were thawed at 4°C and diluted 4-fold in NYCIII media (in an anaerobic chamber). The diluted communities were used for coculture experiments with *F. nucleatum*. Neu5Ac, *N*-acetylneuraminic acid; 4MU, 4-methylumbelliferone.
(TIFF)

**S1 Table. Key bacterial strains, plasmids, and primers.**
(PDF)

**S2 Table. Genetic organization of the putative sialic acid catabolic gene cluster in *F. nucleatum* strain ATCC 23726.** ATCC, American Type Culture Collection.
(PDF)

**S1 Data. Contains underlying numerical data for the graphs in Figs 1–8 and S3–S7 Figs.**
(XLSX)

**S2 Data. Contains underlying data for S3D Fig.**
(XLSX)

## Acknowledgments

We thank Nadum Member-Meneh and Courtney Amegashie for technical contributions to clinical sample collection and processing, Justin Fay for advice on microbiome profiling, Justin Merritt for providing some of the *Fusobacterium* strains, Justin Perry for technical contributions to analysis of sialic acid consumption among *Fusobacteria*, Deborah Frank for editorial assistance and advice, and Marcy Hartstein for assistance with illustrations. We also thank Denise Spear, WHNP-BC, and Valerie Higginbotham, WHNP-BC, for their participation in collecting vaginal swabs from participants and the St. Louis County Department of Public Health for facilitating this study. Finally, we thank the women who have generously provided their informed consent and vaginal samples for use in these studies.

## Author Contributions

**Conceptualization:** Kavita Agarwal, Lloyd S. Robinson, Somya Aggarwal, Andrew L. Kau, Warren G. Lewis, Amanda L. Lewis.

**Data curation:** Kavita Agarwal, Lloyd S. Robinson, Somya Aggarwal, Lynne R. Foster, Ariel Hernandez-Leyva, Hueylie Lin, Brett A. Tortelli, Valerie P. O'Brien, Liza Miller, Hilary Reno.

**Formal analysis:** Kavita Agarwal, Lloyd S. Robinson, Somya Aggarwal, Ariel Hernandez-Leyva, Hueylie Lin, Brett A. Tortelli, Valerie P. O'Brien, Andrew L. Kau, Warren G. Lewis, Amanda L. Lewis.

**Funding acquisition:** Kavita Agarwal, Brett A. Tortelli, Andrew L. Kau, Nicole M. Gilbert, Warren G. Lewis, Amanda L. Lewis.

**Investigation:** Kavita Agarwal, Lloyd S. Robinson, Somya Aggarwal, Lynne R. Foster, Ariel Hernandez-Leyva, Hueylie Lin, Brett A. Tortelli, Valerie P. O'Brien, Liza Miller, Andrew L. Kau, Warren G. Lewis, Amanda L. Lewis.

**Methodology:** Kavita Agarwal, Lloyd S. Robinson, Somya Aggarwal, Lynne R. Foster, Ariel Hernandez-Leyva, Brett A. Tortelli, Valerie P. O'Brien, Liza Miller, Andrew L. Kau, Nicole M. Gilbert, Warren G. Lewis, Amanda L. Lewis.

**Project administration:** Warren G. Lewis, Amanda L. Lewis.

**Resources:** Amanda L. Lewis.

**Software:** Amanda L. Lewis.

**Supervision:** Andrew L. Kau, Hilary Reno, Nicole M. Gilbert, Warren G. Lewis, Amanda L. Lewis.

**Validation:** Kavita Agarwal, Lloyd S. Robinson, Somya Aggarwal, Lynne R. Foster, Hueylie Lin, Brett A. Tortelli, Valerie P. O'Brien, Liza Miller, Andrew L. Kau, Warren G. Lewis, Amanda L. Lewis.

**Visualization:** Kavita Agarwal, Lloyd S. Robinson, Somya Aggarwal, Valerie P. O'Brien, Warren G. Lewis, Amanda L. Lewis.

**Writing – original draft:** Kavita Agarwal, Lloyd S. Robinson, Lynne R. Foster, Nicole M. Gilbert, Warren G. Lewis, Amanda L. Lewis.

**Writing – review & editing:** Kavita Agarwal, Lloyd S. Robinson, Somya Aggarwal, Lynne R. Foster, Ariel Hernandez-Leyva, Hueylie Lin, Brett A. Tortelli, Valerie P. O'Brien, Liza Miller, Andrew L. Kau, Hilary Reno, Nicole M. Gilbert, Warren G. Lewis, Amanda L. Lewis.

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
