## [Editor Report · Decision Letter 0]

29 Jan 2020

Dear Dr Lewis, 

Thank you for submitting your manuscript entitled "Glycan cross-feeding supports mutualism between Fusobacterium and the vaginal microbiota" for consideration as a Research Article by PLOS Biology.

Your manuscript has now been evaluated by the PLOS Biology editorial staff as well as by an academic editor with relevant expertise and I am writing to let you know that we would like to send your submission out for external peer review.

Please re-submit your manuscript within two working days, i.e. by Jan 31 2020 11:59PM.

Kind regards,

Lauren A Richardson, Ph.D

Senior Editor

PLOS Biology

---

## [Decision Letter · Decision Letter 1]

9 Apr 2020

Dear Dr Lewis,

Thank you very much for submitting your manuscript entitled "Glycan cross-feeding supports mutualism between Fusobacterium and the vaginal microbiota" for consideration as a Research Article at PLOS Biology. Thank you also for your patience as we completed our editorial process, and please accept my sincere apologies for the long delay in providing you with our decision. Your manuscript has been evaluated by the PLOS Biology editors, an Academic Editor with relevant expertise, and by three independent reviewers.

As you will see, the reviewers find your conclusions novel and interesting, but also raise several issues that need to be clarified. Reviewer 2 would like know if you have measured the level of free Neu5Ac in the vagina washes to see if it correlates with the increase of the presence of F. nucleatum. After consulting with the academic editor, we do feel that these measurements should be added to the manuscript, while the other points can be addresed by textual changes.

In light of the reviews (attached below), we are pleased to offer you the opportunity to address the comments from the reviewers in a revised version that we anticipate should not take you very long. We will then assess your revised manuscript and your response to the reviewers' comments and we may consult the reviewers again.

We expect to receive your revised manuscript within 2 month, but please email us (plosbiology@plos.org) if you have any questions or concerns, or would like to request an extension - we completely understand that due to COVID-19 your lab might be close and it will take a while for you to be able to add the data requested. At this stage, your manuscript remains formally under active consideration at our journal; please notify us by email if you do not intend to submit a revision so that we may end consideration of the manuscript at PLOS Biology.

**IMPORTANT - SUBMITTING YOUR REVISION**

*Resubmission Checklist*

*Published Peer Review*

*PLOS Data Policy*

*Blot and Gel Data Policy*

Sincerely,

Ines

--

Ines Alvarez-Garcia, PhD

Senior Editor

PLOS Biology

Carlyle House, Carlyle Road

Cambridge, CB4 3DN

+44 1223–446970

Reviewers’ comments

Rev. 1:

In their manuscript “Glycan cross-feeding supports mutualism between Fusobacterium and the vaginal microbiota”, Agarwal et al set out to validate the hypothesis that the increased risk for pathogenic colonization observed in bacterial vaginosis patients is due (at least in part) to the increased presence of sialidases. Specifically, the authors look at the pathogen Fusobacterium nucleatum, which had been predicted to be able to catabolize free sialic-acid but does not possess sialidases. They generate a knockout (siaT) that is lacking key metabolic genes that block the ability of F. nucleatum to catabolize and grow in the presence of free sialic acid, which is then used as a control for the next experiments to distinguish between effects caused by the presence versus catabolic activity of F. nucleatum. Overall Agarwal et al present compelling evidence that certain strains of F.nucleatum can grow on free sialic acid and are thereby provided an advantage in environments where sialidases are present. However, there are some points in the manuscript that need clarification as detailed below.

Major Comments

- The murine in vivo results show that only the WT, not siaT mutant, increase sialidase activity over time (Fig 5B+C). First, it is not clear if the siaT mutant increases sialidase activity compared to no infection control (albeit less than WT). Second, the murine ex vivo results show that both WT and siaT mutants increase sialidase activity in the vaginal microbial community. What accounts for this discrepancy between the in vivo and ex vivo data? This raises major concerns how translational these findings are.

- In the human vaginal swab ex vivo data, only the WT F. nucleatum strain is used, I assume that is because there was a limited amount of sample but it would be interesting to see if the mutant results here reflect the in vivo or ex vivo results from the murine experiments. Additional experiments would help to address this question.

Minor Comments

- Figure 6;

 o Fig6; as written the experiment is not clear; frozen vaginal microbial communities were grown overnight and then mixed with an F. nucleatum strain to test for effect – for how long were they grown again? It took me awhile to find this in Fig S1 (16hrs), I would recommend mentioning this in the results for improved clarity.

 o Fig 6C shows the final F. nucleatum CFU count after incubation with the vaginal pools, but how much F. nucleatum was put in to start? Is this concentration-dependent? It would be nice to see that the amount added was similar between both strains at the beginning as well. 

 o The CFU input should be listed in Fig 6D (at a minimum in the legend if not on the figure) since right now it just shows qualitative increased input with triangles – is the increasing input different by 2-fold? 10-fold? 0.1-fold??

- Fig S4; Even though there is no difference between WT and KO colonization in Jackson mice, when comparing numbers to Envigo mice in Fig 4 it appears that the KO is better at colonizing in Jackson mice rather than the WT being worse (as I would expect since Jackson mice have low sialidase activity) – maybe colonization in these mice is driven by different factors? I think it would be worthwhile to comment on this at least in the supplementary info.

- Lines 260-270; the way this is currently written it sounds like sialidase activity was somehow determined from the extracted DNA – clarifying that 16S rRNA gene amplicon sequencing was performed and in parallel that isolated colonies were screened for sialidase activity will improve the readability of this section.

- Line 273; ‘somewhat different’ is not very descriptive, from Sup Fig3D it appears as though the communities are quite distinct, a PERMANOVA analysis would help strengthen this statement in the text .

- Line289; clarify to lower CFU titers.

- Fig5B; difficult to tell which groups the stars are signifying significance for, use bars like in 5C.

- Fig S5; E.coli misspelled in legend (E.colii).

Rev. 2:

The manuscript reports the characterisation of sialic acid consumption by Fusobacterium nucleatum strains and the impact on sialic acid cross-feeding on F. nucleatum colonisation of the vaginal niche. Specific outputs include the characterisation of NanA (sialic acid lyase) and NanT (sialic acid transporter) as critical components of F. nucleatum sialic acid metabolism in vitro, the ability of F. nucleatum to benefit from a sialidase-rich vaginal microbiota and the impact of F. nucleatum on G. vaginalis. The methodology is sound, and the manuscript presents some very interesting and novel insights into the interplay between sialic acid catabolism and the vaginal microbiota.

Below some comments and suggestions to clarify some aspects of the manuscript.

In the first set of experiments, the authors compared the Neu5Ac levels of F. nucleatum strains grown on (rich?) growth medium as compared to un-inoculated medium but don't provide information (in Results or in Fig legend) on the composition of the medium and whether it was supplemented with Neu5Ac (using the semi-defined medium introduced on p. 8. This information should be provided. The analysis uses un-inoculated medium as a control, a more informative control would be to use medium with no Neu5Ac as carbohydrate source. The authors should also include in the Results section how the Neu5Ac concentrations were determined (p. 5), and how they measured free versus bound sialic acid (p.11).

The authors demonstrated that in vitro F. nucleatum is not able to utilise a synthetic sialylated substrate and that adding exogenous sialidase from Arthobacter ureafaciens enabled growth of F. nucleatum on this substrate. The interpretation of the data is based on growth measurements and supporting data such as measuring the Neu5Ac in the medium and/or the sialidase activity during growth would be needed to strengthen the interpretation. Also, the authors may comment why G. vaginalis and not Arthobacter ureafaciens was used as a control in the first growth experiment (L.213) or conversely why the sialidase from G. vaginalis was not used in the following experiment (from L.218). The authors should also provide information on the linkage of the substrate as differences in sialidase specificity of the bacteria tested will need to be considered when interpreting the results. Did the authors try and co-culture a sialidase-producing bacteria from the vaginal niche (e.g. G. vaginalis, Arthobacter ureafaciens or Enterococcus casseliflavus/gallinarum) with F. nucleatum on sialylated substrate?

With regards to the mouse work, the authors demonstrate a link between vaginal microbial sialidase activity and F. nucleatum colonisation. It is suprising that the authors did not measure the level of free Neu5Ac in the vaginal washes of the mice under study so to link the increase in F. nucleatum with increased availability of free Neu5Ac in mice displaying high sialidase activity. Did/could the authors carry out these investigations? These data would need to be included to be consistent with the title of the manuscript that glycan cross-feeding supports mutualism between Fusobacterium and the vaginal microbiota.

The last part of the study revealing an increase in microbial sialidase activity of mouse or human vaginal communities in the presence of F. nucleatum is interesting but, is more of an observation, and more work would be needed to strengthen this part of the work, which may be outsisde the scope of this manuscript. I would therefore suggest removing these last two sections from the manuscript as without further supporting data, the possible explanations provided are too speculative at this stage.

Minor comments.

The sentences L. 127 and L211 about the bioinformatics analysis of F. nucleatum 'proteomes' are ambiguous, as this analysis is based on genome screening not on proteomics data.

Fig. 1 Legend requires clarification with regards to the growth medium used in the experiment and as control. 

L.186,187. 'plateauing' sounds like of Lab jargon and may be replaced by 'reaching a plateau'

L. 274 the term used 'somewhat different' vaginal microbiota is too vague and should be refined.

Rev. 3: Carl Yeoman – please note that this reviewer has waived anonymity

Apologies for the delay, I really enjoyed reading this paper and felt it was another great and useful addition to the literature from the Gilbert lab. The manuscript describes a number of well designed and deployed experiments that examine the microbial ecology related to Fusobacterium nucleatum colonization and persistence of the vaginal microenvironment and a potential mutualism with G. vaginalis adding important insight into the ecological interactions underlying vaginal dysbiosis and related-diseases, with particular relevance to pre-term birth. I commend the authors on the amount of work that has clearly gone into this manuscript and the well thought out set of experiments. Below are only minor thoughts and comments for consideration:

Minor Comments:

Line 54: Beware of the creep in reporting of these statistical data - available data supports a little less than 30 % of US women are affected by BV, which has since been reported as 30 % and subsequently as a third of US women. This may seem small but is actually very large. 

Lines 62 - 64: Aside from the grammatical error of placing the comma (which should probably be a semicolon) inside the brackets, this sentence appears to be miswritten - the healthy microbiome excluding potentially pathogenic members leads to dysbiosis?

Lines 481+ In addition to oro-vaginal contact, other routes have previously been proposed, including in the context of pre-term delivery (e.g. see Cobb et al. 2017. DOI: 10.2147/IJWH.S142730) and while I don't think evidence supports a placental microbiome, of relevance to this paper is the similarities 16S studies have found between the placenta and oral microbiota (e.g. Aagaard et al. 2016. DOI: 10.1126/scitranslmed.3008599).

---

## [Editor Report · Decision Letter 2]

29 May 2020

Dear Dr Lewis,

Thank you for submitting your revised Research Article entitled "Glycan cross-feeding supports mutualism between Fusobacterium and the vaginal microbiota" for publication in PLOS Biology. I have now discussed your revision with the team of editors and obtained advice from the Academic Editor.

We're delighted to let you know that we're now editorially satisfied with your manuscript. However before we can formally accept your paper and consider it "in press", we also need to ensure that your article conforms to our guidelines. A member of our team will be in touch shortly with a set of requests. As we can't proceed until these requirements are met, your swift response will help prevent delays to publication. Please also make sure to address the data and other policy-related requests noted at the end of this email.

*Copyediting*

*Published Peer Review History*

*Early Version*

*Submitting Your Revision*

Sincerely,

Ines

--

Ines Alvarez-Garcia, PhD

Senior Editor

PLOS Biology

Carlyle House, Carlyle Road

Cambridge, CB4 3DN

+44 1223–442810

Fig. 1A, C; Fig. 2C, D; Fig. 3A, D; Fig. 4B, C, D; Fig. 5A, B, C, Fig. 6B, C, D; Fig. 7B, C, D; Fig. 8A, B, C; Fig. S3A, B, C, F; Fig. S4A, B; Fig. S5A, B; Fig. S6A, B and Fig. S7

---

## [Editor Report · Decision Letter 3]

10 Jul 2020

Dear Dr Lewis,

On behalf of my colleagues and the Academic Editor, Ken Cadwell, I am pleased to inform you that we will be delighted to publish your Research Article in PLOS Biology. 

Early Version

PRESS 

Kind regards,

Alice Musson

Publishing Editor, 

PLOS Biology

on behalf of

Ines Alvarez-Garcia,

Senior Editor

PLOS Biology